# Disentangling Hyperedges through the Lens of Category Theory

**Yoonho Lee**
KAIST
smlo399benbm@kaist.ac.kr

**Junseok Lee**
KAIST
junseoklee@kaist.ac.kr

**Sangwoo Seo**
KAIST
sangwooseo@kaist.ac.kr

**Sungwon Kim**
KAIST
swkim@kaist.ac.kr

**Yeongmin Kim**
KAIST
cytotoxicity8@kaist.ac.kr

**Chanyoung Park**[*]
KAIST
cy.park@kaist.ac.kr

## Abstract

Despite the promising results of disentangled representation learning in discovering latent patterns in graph-structured data, few studies have explored disentanglement for hypergraph-structured data. Integrating hyperedge disentanglement into hypergraph neural networks enables models to leverage hidden hyperedge semantics, such as unannotated relations between nodes, that are associated with labels. This paper presents an analysis of hyperedge disentanglement from a category-theoretical perspective and proposes a novel criterion for disentanglement derived from the naturality condition. Our proof-of-concept model experimentally showed the potential of the proposed criterion by successfully capturing functional relations of genes (nodes) in genetic pathways (hyperedges). Our implementation is available at https://github.com/Yoonho-Lee-AI4Science/Natural-HNN.

## 1 Introduction

Disentangled representation learning, which aims to identify underlying factors behind observed data, has been applied to graph neural networks (GNNs) to capture hidden semantics or mechanisms in graph-structured data. In molecular graphs, for example, molecular properties are determined by underlying graph-level mechanisms, where specific substructures play distinct roles in shaping these properties. To reflect such graph-level mechanisms, graph-level disentanglement can be designed to capture multiple substructures, each corresponding to different molecular properties. As another example, in opinion dynamics [54, 29, 28], which studies how individuals' opinions evolve through interactions within a social network, an individual's opinion can change after engaging in discussions with neighbors. These discussions act as edge-level mechanisms that influence opinion updates. To reflect edge-level mechanisms, edge-level disentanglement can be designed to capture multiple topics underlying discussions, each affecting different aspects of individual opinions. Depending on the type of mechanism, several types of disentanglement, including node-level [47], edge-level [85], and graph-structure-level [77] approaches, have been proposed.

A fundamental challenge lies in designing a criterion for disentanglement, which determines how relevant each factor is to each mechanism (e.g., each node, edge, or subgraph). Since the representation reflects each factor in proportion to its relevance determined by the criterion, the criterion should be designed in accordance with the type of disentanglement to ensure that the intended type of mechanism is properly captured in the representation. Thus, many disentanglement models strive to identify fundamental characteristics associated with the type of disentanglement and incorporate them into the design of the criterion.

---

[*]Corresponding author.

39th Conference on Neural Information Processing Systems (NeurIPS 2025).

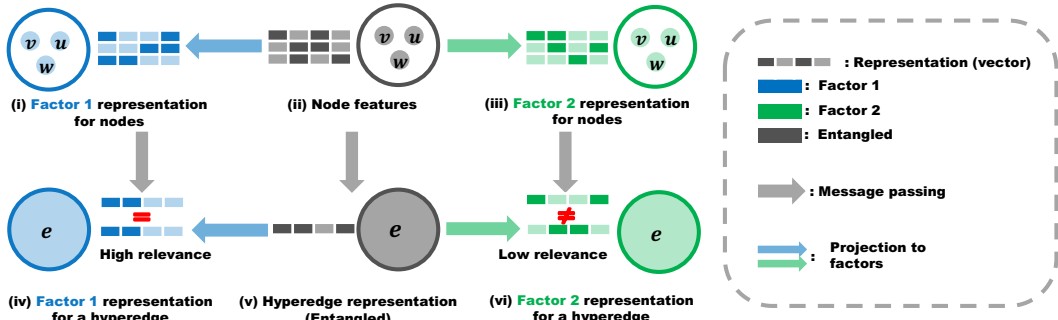

Figure 1: The factor representation consistency criterion assigns a high relevance score when the factor representation learned by two different routes is similar (i.e., consistent factor representation).

Despite numerous studies on disentanglement conducted so far [47, 85, 77, 31, 51], hyperedge disentanglement remains largely unexplored. Hyperedge disentanglement assumes that group interactions (i.e., hyperedges) have mechanisms that determine the labels, and aims to capture the factors (i.e., context or condition) that influence these group interactions. A representative example is the genetic pathway, which is a set of genes (i.e., a hyperedge) that interact to perform a specific biological function. When a pathway becomes dysregulated, its associated biological function can be impaired, potentially leading to diseases such as cancer. The functional context of a pathway can thus be regarded as an underlying factor that governs how group interaction among genes influences high-level labels, such as disease types [65, 74]. Therefore, we aim to design a criterion for hyperedge-level disentanglement that enables a model to capture hyperedge-level factors, such as the functional context of genetic pathways.

To the best of our knowledge, we are the first to propose a criterion designed for hyperedge disentanglement. To identify characteristics that can be derived from the definition of hyperedge disentanglement, rather than from any data-specific assumptions, we analyzed the hypergraph message passing neural network (MPNN) and hyperedge disentanglement from a category-theoretical perspective, as this viewpoint provides a global structural understanding of 'how the system works.' We discovered that the naturality condition holds between entangled and disentangled representations, and we used this as a characteristic associated with hyperedge disentanglement. Based on this characteristic, we defined factor representation consistency as the criterion. Figure 1 briefly illustrates our criterion for hyperedge disentanglement. As shown in Figure 1, there are two ways to obtain hyperedge factor representations: one by disentangling first and then performing message passing (i.e., $(ii) \rightarrow (i) \rightarrow (iv)$), and the other by performing message passing first and then disentangling (i.e., $(ii) \rightarrow (v) \rightarrow (iv)$). Our criterion suggests that the hyperedge factor representation learned by both methods should be similar (i.e., consistent representation) when the factor is relevant to hyperedge disentanglement. To validate whether our novel criterion can disentangle hyperedges, we created a *proof-of-concept* model, **Natural-HNN** (Naturality-guided disentangled Hypergraph Neural Network), and performed a cancer subtype classification task with hypergraphs of genetic pathways. Our model outperformed the baselines by successfully capturing the functional context of pathways, which are the underlying factors influencing group interactions.

Our main contributions are summarized as follows:

- This paper, for the first time, provides an analysis of hypergraph message passing neural networks and hyperedge disentanglement through the lens of category theory.
- Based on the analysis, we derive a novel criterion for hyperedge disentanglement. To the best of our knowledge, this is the first paper to propose a criterion for hyperedge disentanglement.
- We create a simple yet effective *proof-of-concept* model, Natural-HNN, and performed a cancer subtype classification task. Experimental results showed that the model could capture the functional context of pathways, which are factors associated with hyperedge disentanglement.

## 2 Related Work

In Section 2.1, we briefly describe a criterion widely used in disentangled representation learning and discuss why it may not be suitable for hyperedge disentanglement. In Section 2.2, we discuss how category theory has been applied in the field of deep learning and explain how we adopt the theory to our problem at hand. In Section 2.3, we briefly summarize several hypergraph neural networks.

## 2.1 Disentangled Representation Learning (DRL)

Disentangled representation learning consists of three components: factor encoder, factor discrimination loss, and criterion. A factor encoder projects the entangled representation into factor-specific representations. These factor encoders are implemented as $K$ MLPs, where $K$ denotes the number of factors, given as a hyperparameter. To encourage each factor representation to contain distinct information, factor discrimination losses, including factor classifier-based loss [85], factor-wise contrastive learning loss [41], and the Hilbert-Schmidt Independence Criterion (HSIC) [47], are used.

The disentanglement criterion is the most crucial component of DRL, as it determines the relevance of each factor and consequently how much it is reflected in the representation. This criterion is designed based on the characteristics that the intended disentangled factors should ideally possess. Although defining the criterion based on such ideal properties does not theoretically guarantee successful disentanglement, numerous studies have empirically confirmed that these criteria indeed enable effective disentanglement. For example, in the early image generative models that pioneered DRL, disentanglement was guided by adopting the equivariant property as the disentanglement criterion [30], since an ideal generative factor should cause the image to vary equivariantly with changes in the generative factor. Therefore, many studies have strived to identify suitable ideal characteristics for the type of disentanglement they pursue, under the assumption that such ideal properties of factors facilitate effective disentanglement.

In the field of graph and hypergraph representation learning, disentanglement has been used to exploit hidden semantics behind subgraphs or interactions with neighbors. Since such hidden semantics are highly abstract concepts, it is difficult to identify generally applicable properties. Consequently, as these semantics exhibit different characteristics depending on the data, the design of criteria has often relied heavily on assumptions about the data. The most widely used criterion for disentanglement is the factor representation similarity-based approach. For example, DisenGCN [49] assumes that the $k$-th factor is likely the reason behind the existence of an edge in a graph if the $k$-th factor representations of the two connected nodes are similar. A similar criterion is also used by HSDN [31], which performs hypergraph-structure-level disentanglement aimed at identifying substructures that contribute to hypergraph properties. The authors of the paper assume that important hyperedges would share commonalities and therefore need to have similarity in the factor representations of nodes.

However, factor representation similarity-based criterion may not be suitable for hyperedge disentanglement because the way group interactions influence labels are not necessarily related to the similarity or commonalities between participants. For instance, consider the case of opinion dynamics involving a group engaged in a discussion; the topic of such discourse need not necessarily pertain to the commonalities shared among its participants. One can easily imagine a situation where researchers from diverse fields gather to discuss and solve complex and challenging problems. As another example, in genetic pathways, the similarity of gene expression values (i.e., gene features) of the constituent genes bears no relation to the functional context. Therefore, since the existing criteria based on data-specific assumptions are not suitable for hyperedge disentanglement, we aim to develop a broadly applicable hyperedge disentanglement criterion that does not rely on heuristics.

To develop a universally applicable criterion that does not rely on heuristics or data-specific assumptions, we first need to analyze how hidden semantics are involved in the mechanisms through which group interactions contribute to labels, and to derive the corresponding characteristics or properties from this analysis. However, since the hidden semantics underlying group interactions are highly abstract concepts, conducting such an analysis is inherently challenging. To address this challenge, we employ category theory, which is well-suited for representing and analyzing systems as compositional structures. By formulating hyperedge disentanglement and investigating how factors contribute to the label-mapping mechanism through the lens of category theory, we discovered that a naturality condition must hold between the entangled and disentangled representations of nodes and hyperedges. Based on this observation, we derive a novel criterion based on hyperedge representation consistency. Finally, we conclude this section with a formal definition of hyperedge disentanglement.

**Hyperedge Disentanglement.** A hypergraph with $N$ nodes and $M$ hyperedges can be represented by incidence matrix $\mathcal{I} \in \{0,1\}^{N \times M}$, which indicates whether a node belongs to a hyperedge or not. Hyperedge disentanglement assumes the existence of multiple hidden factors underlying group interactions, which influence how labels are determined, and aims to capture these factors while predicting the labels. In other words, it is assumed that there exists a set of disentangled incidence

matrices $\mathcal{I}^{dis} = \{\mathcal{I}_1, \ldots, \mathcal{I}_K\}$ which are not explicitly provided in the data, where $\mathcal{I}_i$ denotes incidence matrix of subhypergraph for factor $i$. The objective of hyperedge disentanglement is to learn a hypergraph neural network $f_{HNN}(\mathcal{I}, X)$ that approximates the ground-truth label mapping function $f_{data}(\mathcal{I}^{dis}, X)$ by learning an approximation of $\mathcal{I}^{dis}$. Note that hyperedge-level disentanglement differs from hypergraph structure-level disentanglement, as the latter assumes that the presence of certain substructures determines the labels (i.e., $f_{data}(\mathcal{I}^{dis})$).

## 2.2 Category Theory for Deep Learning

Category theory is an abstract language of mathematics that focuses on the compositional structure of a system. One of the applications in the field of deep learning that uses category theory is neural algorithmic reasoning [68] which aims to train a neural network that can execute algorithmic computation in latent space. Several studies [14, 15] have attempted to align the computational structure of an algorithm with that of the model to effectively approximate computer algorithms. The motivation for aligning the structures comes from the theoretical conclusion [75] that structurally aligned models generalize better due to lower sample complexity (i.e., they require fewer samples in training to ensure low test error). Motivated from the works above, we analyze a hyperedge disentanglement model using category theory from the perspective that the computational structure of the model should be structurally aligned with the factor-related mechanism. Through this formulation, we identify a characteristic that can serve as a criterion. Note that the basic concepts in category theory we used are described in Appendix A.

## 2.3 Hypergraph Neural Networks (HNNs)

Several HNN models have been recently proposed to leverage information contained in multiway interaction. HGNN [20] and HCHA [3] use a normalized hypergraph Laplacian, which is mathematically equivalent to clique expansion (CE) [67], and apply the traditional graph convolution mechanism. HNHN [12] additionally adopts nonlinearity when calculating hyperedge representations to differentiate a hypergraph from a clique expanded graph, while UniGNN [32] unifies HNNs and GNNs into the same framework. Moreover, HyperGAT [11] adopts the attention mechanism to HNN for text classification, and SHINE [48] proposes dual attention mechanism for the disease classification task. ED-HNN [70] proposes equivariant message passing HNN, which allows hyperedges to propagate different messages to its incident nodes. AllDeepSets and AllSetTransformer [6] consider a hyperedge as a set and apply DeepSets [83] and Set Transformer [37], respectively, to increase expressive power of HNN.

Efforts to apply disentanglement to hypergraph-structured data have been relatively limited. HIDE [42] and DisenHCN [43] applied hypergraph disentanglement in the context of recommender systems. However, in these works, the hyperedge semantics were explicitly provided as hyperedge types in the data, and their approaches focused on disentangling node features corresponding to each hyperedge type, rather than capturing the underlying hyperedge semantics. HSDN [31] proposed hypergraph structure-level disentanglement, rather than hyperedge-level disentanglement.

## 3 Categorical Interpretation of Message Passing HNN and disentanglement

Before addressing hyperedge disentanglement, we first analyze the relationship between the mechanism by which group interactions influence labels and hypergraph MPNNs from the perspective of category theory, which will be discussed in Section 3.1. In Section 3.2, we further concretize this analysis by examining how factors relate to the mechanism and describe the process of deriving the characteristic (i.e., naturality condition) from it.

**Notation.** Let $\mathcal{G} = (\mathcal{V}, \mathcal{E})$ denote a hypergraph, where $\mathcal{V} = \{v_1, v_2, ..., v_N\}$ indicates a set of nodes and $\mathcal{E} = \{e_1, e_2, ..., e_M\}$ indicates a set of hyperedges, where $N = |\mathcal{V}|$ and $M = |\mathcal{E}|$ are the number of nodes and the number of hyperedges in a hypergraph $\mathcal{G}$, respectively. A set of node features given as input to each layer of the model is denoted as $X = \{x_{v_1}, ..., x_{v_N}\}$, a set of hyperedge representations (calculated in each layer of the model) is denoted as $H = \{h_{e_1}, ..., h_{e_M}\}$, and a set of representations obtained after message passsing is denoted as $Y = \{y_{v_1}, ..., y_{v_N}\}$. '$en$' denotes an entangled object

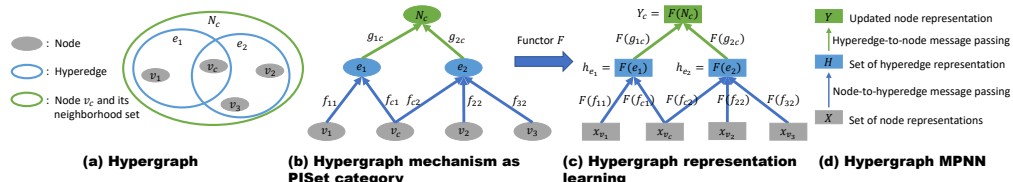

Figure 2: Compositional structure in hypergraph representation learning.

or morphism and is written in superscript or subscript, while '*dis*' denotes a disentangled object or morphism. The symbol '$\mathbin{\text{\textsc{g}}}$' is used to denote the composition of morphisms.[1]

## 3.1 Compositionality in Hypergraph Representation Learning

Most hypergraph representation learning methods produce the representation of a node by integrating its own representation and its neighbors' representations defined by a hypergraph topology. The fundamental assumption underlying these models is that group interactions with neighbors contribute, in some manner, to the labels. To further elucidate this assumption, consider the hypergraph example depicted in Figure 2 (a). Although not given by the hypergraph topology, we introduced the set $N_c$, which includes the center node $v_c$ and its neighbors, in order to represent the information that $v_c$ possess after message passing. Then, the assumption can be illustrated as in Figure 2 (b). Each group interaction, given as a hyperedge, can produce new meanings or information (e.g., a new meaning for $e_1$) through some interaction mechanisms (e.g., $f_{11}, f_{c1}$). Subsequently, the assumption posits that this newly generated information influences the participants (e.g., $v_c$) of the group interaction via some mechanism (e.g., $g_{1c}, g_{2c}$), thereby resulting in new information (e.g., $N_c$) for the participants (e.g., $v_c$) that may be associated with the label.

The abstract description above can be formalized through the lens of category theory. Specifically, if we consider each node as a set, since a hyperedge contains nodes, there are morphisms (inclusion) between nodes and hyperedges induced by the poset structure. We defined this as **PISet**, the category with **p**oset structure where morphisms are **i**nclusions and objects are **set**s. Thus, we can see nodes $(v_1, v_c, v_2, v_3)$ and hyperedges $(e_1, e_2)$ constitute **PISet** as shown in Figure 2 (b), where gray-colored nodes and hyperedges are set objects, and inclusions are morphisms (blue arrow) between sets. The same mechanism holds between hyperedges $(e_1, e_2)$ and a set $N_c$ that includes node $v_c$ and its neighbors. In Figure 2 (b), for instance, we can see hyperedges $(e_1, e_2)$ and $N_c$ constitute **PISet** as they have morphisms (green arrow) induced by the poset structure.

In order to learn and predict with computers, such objects and morphisms must be expressed in numerical values and their transformations. Hence, we define a category of deep learning representations, **DLRep**, where objects are vector representations and morphisms are transformations between them. Figure 2 (c) shows the result of applying a functor $F : \textbf{PISet} \rightarrow \textbf{DLRep}$, which can be simplified to a diagram in Figure 2 (d). Thus, any kind of hypergraph MPNNs can be seen as a way of learning representations and their transformations respecting compositional structure of entities. In other words, hypergraph MPNNs can be seen as structurally aligned, to some extent, with the mechanisms by which group interactions present in hypergraph data influence the labels.

However, the degree to which a model is structurally aligned depends on implementation details. For example, convolution-based models are structurally well-aligned with mechanisms in which all nodes contribute equally during group interactions. Conversely, when node contributions vary within the group interaction, attention-based methods are more structurally aligned with the mechanism than convolution-based ones. Therefore, to perform hyperedge disentanglement, we structurally analyze how factors are involved in the mechanism and, in Section 3.2, investigate the characteristics of a hyperedge disentanglement model that is well structurally aligned with the mechanism.

## 3.2 Guiding Disentanglement with Naturality Condition

Since entangled and disentangled representations are different ways of representing the same compositional structure, we can regard them as the result of applying two different functors $F : \textbf{PISet} \rightarrow \textbf{DLRep}$ (for entangled representations) and $G : \textbf{PISet} \rightarrow \textbf{DLRep}$ (for disentangled representations) as shown in Figure 3 (a). Thus, we have the naturality condition between

---

[1]Two notations $f \mathbin{\text{\textsc{g}}} g$ and $g \circ f$ have the same meaning : "applying $f$ first, and then applying $g$". We use the notation '$\mathbin{\text{\textsc{g}}}$' following [23].

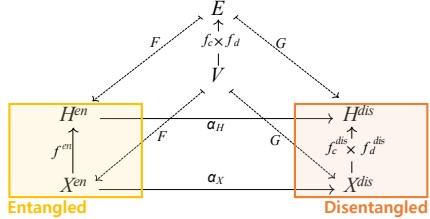
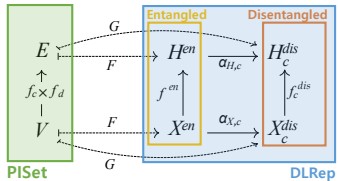

(a) Natural Transformation in HNN        (b) Natural Transformation in HNN, factor perspective

Figure 3: Naturality condition in disentangled representation learning to capture group interaction mechanism related factors. $X$ denotes a set of node representations and $H$ denotes hyperedge representation. $V$ and $E$ denote nodes and hyperedge in **PISet**. '$c$' and '$d$' denotes factors.

entangled and disentangled representations. Figure 3 (b) is equivalent to Figure 3 (a), but only the components related to the factor '$c$' are shown (explanations are in Appendix A.6). Note that $\alpha_{X,c} = \alpha_X \,\mathring{_9}\, p_c$ where $p_c : X^{dis} \to X_c^{dis}$. If factor '$c$' is relevant to the morphism between node set $V$ and hyperedge $E$, the naturality condition must hold for the perspective of factor '$c$'. Thus, factor '$c$' representation of a hyperedge (i.e., $H_c^{dis}$) must be the same (or similar) regardless of applying $f^{en} \,\mathring{_9}\, \alpha_{H,c}$ (i.e., message passing on entangled representation first, and then disentangling factors) or $\alpha_{X,c} \,\mathring{_9}\, f_c^{dis}$ (i.e., disentangling factors first, and then message passing on disentangled representation). In other words, the factor representation must be consistent regardless of the sequence of operations if that factor is relevant to the interaction context of a hyperedge. We use this property as a guidance for disentanglement, since it must hold for any kind of hypergraph message passing neural networks, and must work regardless of data characteristics.

# 4 Proof-of-concept model : Natural-HNN

To validate whether our criterion can effectively capture factors relevant to hyperedge disentanglement, we implemented a simple yet effective model, Natural-HNN. Each layer of the model is consisted with 3 components as shown in Figure 4: **1)** Node-to-hyperedge propagation step that learns hyperedge factor representations and relevance scores, which is calculated by our criterion. **2)** Hyperedge-to-Node propagation step that propagates factor representations of hyperedges to nodes with weights proportional to relevance scores. **3)** The last component concatenates factor representations and produces final outputs by interpolating with the node representations given as input to the layer. Note that each layer of Natural-HNN has $K$ factors where $K$ is a hyperparamter.

## 4.1 Node-to-Hyperedge Factor Propagation

**Obtaining Two Disentangled Hyperedge Representations.** To validate whether the naturality condition (Figure 4 (a)) holds, we need to get two disentangled hyperedge factor representations for every factor (i.e., $H_k^{dis}$ for every factor $k \in [1, K]$). The two disentangled representations are obtained through 1) Aggregation-first Branch and 2) Disentalgle-first Branch. In the following, we describe how morphisms in Figure 4 (a) are implemented as operations in the two branches shown in Figure 4 (b).

- **Aggregation-first Branch.** The first disentangled representation is obtained from the aggregation-first branch performing $f^{en} \,\mathring{_9}\, \alpha_{H,k}$ for each factor $k$. This process is implemented as performing aggregation $agg_{n2e}$ (i.e., $f^{en}$ in Figure 4 (a)) first, and then disentangling into hyperedge factor representations using a factor encoder $\alpha_{H,k}$. The factor representations of hyperedge $e_i$ obtained from this branch are denoted as $\tilde{h}_{e_i}^1, \ldots, \tilde{h}_{e_i}^K$.
- **Disentangle-first Branch.** The other one is obtained from the disentangle-first branch performing $\alpha_{X,k} \,\mathring{_9}\, f_k^{dis}$ for each factor $k$. This process is implemented as disentangling into node factor representations with factor encoder $\alpha_{X,k}$ first, and then performing aggregation $agg_{n2e}$ (i.e., $f_c^{dis}$ in Figure 4 (a)). Factor representations of hyperedge $e_i$ obtained from this branch are denoted as $h_{e_i}^1, \ldots, h_{e_i}^K$.

For both branches, we used mean aggregation as $agg_{n2e}$ and $K$ MLPs as factor encoders for disentangling factors. Factor representations are vectors with size $d/K$ (i.e., $h_{e_i}^k, \tilde{h}_{e_i}^k \in \mathbb{R}^{\frac{d}{K}}$), when the desired size for node representations after message passing is $d$. In summary, operations of the two branches regarding factor $k$ can be written as follows:

$$\tilde{h}_{e_j}^k = \text{MLP}_k(\text{mean}(\{x_{v_i} | v_i \in e_j\})), \quad h_{e_j}^k = \text{mean}(\{\text{MLP}_k(x_{v_i}) | v_i \in e_j\}) \tag{1}$$

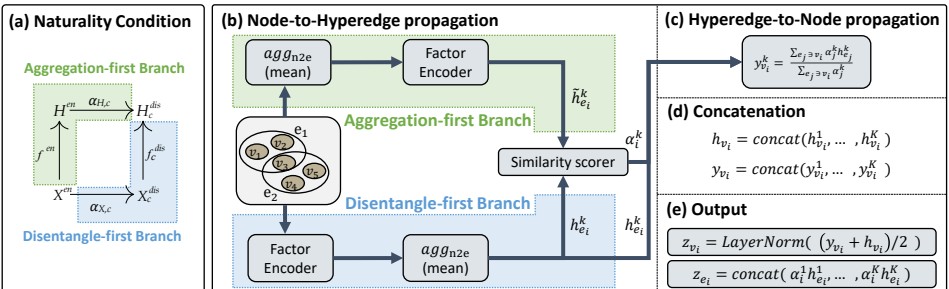

Figure 4: Architecture of proof-of-concept model Natural-HNN. It calculates the relevance of factor $k$ ($\alpha_i^k$) and performs weighted message passing for each factor.

**Deciding Factors with Consistency.** The extent to which the naturality condition is satisfied can be measured by calculating the similarity between the two disentangled hyperedge factor representations $\tilde{h}_{e_j}^k$ and $h_{e_j}^k$. In other words, we can consider that the naturality condition holds when the two representations are similar (i.e., consistent), and does not hold when the two representations are largely different. We introduce a similarity scorer that calculates the similarity of two $L_2$-normalized vectors. Specifically, we calcualte the relevance or importance of factor $k$ for a hyperedge $e_i$ as $\alpha_i^k = \sigma\left(\frac{h_{e_i}^k}{\|h_{e_i}^k\|_2} W_k \frac{\tilde{h}_{e_i}^{k^T}}{\|\tilde{h}_{e_i}^k\|_2}\right)$, where $W_k \in \mathbb{R}^{\frac{d}{K} \times \frac{d}{K}}$ is a learnable parameter matrix for factor $k$, and $\sigma$ is the sigmoid function. Lastly, we obtain the final hyperedge factor representations by multiplying $\alpha_i^k$ to the corresponding hyperedge factor representations obtained from the disentangle-first branch[2], i.e., $\alpha_i^k h_{e_i}^k$, that reflects the relevance of the factor $k$ for the hyperedge $e_i$.

## 4.2 Hyperedge-to-Node Factor Propagation

When aggregating hyperedge representations (i.e., $\alpha_i^k h_{e_i}^k$) to update node representations, the sum of neighboring hyperedge representations with respect to factor $k$ must be divided by the sum of $\alpha_i^k$ so that hyperedge relevance scores (i.e., $\alpha_i^k$) are normalized during aggregation. Thus, the updated factor $k$ representation of node $v_i$, i.e., $y_{v_i}^k$, can be written as $y_{v_i}^k = \frac{1}{\sum_{e_j \ni v_i} \alpha_j^k} \sum_{e_j \ni v_i} \alpha_j^k h_{e_j}^k$.

## 4.3 Final Output of each Layer of Natural-HNN

To allow a model to determine its focus between information from neighbors (i.e., $y_{v_i}$) and information from the node itself (i.e., $x_{v_i}$), one can introduce a hyperparameter $\beta$ that determines the interpolation ratio between them (i.e., interpolate in $\beta : 1 - \beta$ ratio). However, for simplicity, we set $\beta = 0.5$, so that the two pieces of information are interpolated in a 1:1 ratio. To make sure that interpolation is performed on disentangled representations, we used the factor encoder used in the message passing step (i.e., $h_{v_i}^k = \text{MLP}_k(x_{v_i})$). Specifically, $z_{v_i} = \text{LayerNorm}(0.5 y_{v_i} + 0.5 h_{v_i})$, where $y_{v_i} = \text{Concat}(y_{v_i}^1, \ldots, y_{v_i}^K)$, $h_{v_i} = \text{Concat}(h_{v_i}^1, \ldots, h_{v_i}^K)$.

## 4.4 Optional: Factor Discrimination Loss

Existing disentangled representation learning methods [47, 77] have widely adopted a factor discrimination loss aiming at promoting factors to contain different information. Following [85], we added a factor discrimination loss $\mathcal{L}_{dis}$ to the final loss, i.e., $\mathcal{L} = \mathcal{L}_{task} + \lambda \mathcal{L}_{dis}$[3], where $\lambda$ is a factor discrimination loss weight given as a hyperparameter. Details can be found in the Appendix C.2. Using the factor discrimination loss increases the performance of our model (Table 7) and helps each factor to contain different information (Figure 6). However, introducing this loss requires additional hyperparameter tuning for $\lambda$, which often involves a large search space and increases experimental runtime. Considering that this loss is not closely related to our primary experimental objective—validating whether our proposed criterion captures factors relevant to hyperedge disentanglement—**we consider it an optional component** of the *proof-of-concept* model.

---

[2]Although we choose the disentangle-first branch here, we can instead use the output of the aggregation-first branch. Both choices give similar results. Please refer to Appendix E.1.

[3]$\mathcal{L}_{task}$ denotes the task related loss calculated from cross-entropy loss with labels and predictions. Details are available at Appendix C.3

Table 1: Model performance on cancer subtype classification task (Macro F1). Top two models are colored by **First**, **Second**. † : the variant of the model using multihead attention. ⋆ : $\mathcal{L}_{dis}$ is not used.

| Method | BRCA | STAD | SARC | LGG | HNSC | CESC |
|---|---|---|---|---|---|---|
| HGNN | 0.726 ± 0.053 | 0.563 ± 0.040 | 0.684 ± 0.067 | 0.694 ± 0.033 | 0.799 ± 0.053 | 0.835 ± 0.052 |
| HCHA | 0.704 ± 0.051 | 0.558 ± 0.044 | 0.675 ± 0.068 | 0.682 ± 0.041 | 0.783 ± 0.055 | 0.844 ± 0.054 |
| HNHN | 0.697 ± 0.046 | 0.573 ± 0.072 | 0.688 ± 0.075 | 0.674 ± 0.038 | 0.791 ± 0.035 | 0.837 ± 0.059 |
| UniGCNII | 0.697 ± 0.052 | 0.617 ± 0.059 | **0.728** ± 0.066 | 0.663 ± 0.039 | 0.830 ± 0.030 | 0.841 ± 0.046 |
| AllDeepSets | 0.716 ± 0.058 | 0.557 ± 0.044 | 0.599 ± 0.058 | 0.665 ± 0.046 | 0.801 ± 0.058 | 0.870 ± 0.044 |
| AllSetTransformer | 0.743 ± 0.057 | 0.553 ± 0.046 | 0.719 ± 0.052 | 0.653 ± 0.038 | 0.814 ± 0.036 | 0.847 ± 0.046 |
| HyperGAT | 0.637 ± 0.121 | 0.534 ± 0.063 | 0.574 ± 0.153 | 0.665 ± 0.054 | 0.789 ± 0.061 | 0.832 ± 0.046 |
| HyperGAT† | 0.641 ± 0.115 | 0.502 ± 0.087 | 0.584 ± 0.150 | 0.646 ± 0.043 | 0.791 ± 0.079 | 0.827 ± 0.041 |
| SHINE | 0.446 ± 0.155 | 0.371 ± 0.135 | 0.529 ± 0.160 | 0.628 ± 0.104 | 0.718 ± 0.055 | 0.745 ± 0.159 |
| SHINE† | 0.651 ± 0.053 | 0.532 ± 0.064 | 0.673 ± 0.059 | 0.650 ± 0.046 | 0.770 ± 0.040 | 0.837 ± 0.061 |
| HSDN | **0.757** ± 0.044 | **0.629** ± 0.045 | 0.726 ± 0.045 | 0.692 ± 0.038 | 0.811 ± 0.044 | 0.867 ± 0.033 |
| ED-HNN | 0.735 ± 0.047 | 0.615 ± 0.050 | 0.718 ± 0.071 | **0.700** ± 0.030 | 0.835 ± 0.047 | 0.875 ± 0.053 |
| ED-HNNII | 0.722 ± 0.045 | 0.536 ± 0.057 | 0.650 ± 0.087 | 0.695 ± 0.039 | **0.845** ± 0.025 | **0.895** ± 0.044 |
| Natural-HNN⋆ (Ours) | **0.804** ± 0.036 | **0.659** ± 0.049 | **0.745** ± 0.045 | **0.707** ± 0.035 | **0.862** ± 0.045 | **0.881** ± 0.042 |

## 5 Experiment

To evaluate our criterion, we performed a cancer subtype classification task from genetic pathways using our *proof-of-concept* model, Natural-HNN. Genetic pathways possess unannotated or hidden functional contexts (i.e., factors) underlying group interactions. Since these are closely linked to cancer and disease, they serve as appropriate data for validating the criterion. Through experiments, we aim to answer the following questions:

- **RQ1** Does Natural-HNN perform well on data where factors, such as functional context, underlying the mechanism are present? (Section 5.2)
- **RQ2** Are the factors captured by Natural-HNN related to hyperedge disentanglement? In other words, are they related to the functional context? (Section 5.3)
- **RQ3** Can Natural-HNN generalize well? And how much is Natural-HNN affected by hyperparameters? (Section 5.4)

### 5.1 Experimental Setup

**Dataset.** For the cancer subtype classification task, we downloaded clinical data for 6 cancer types (BRCA, STAD, SARC, LGG, CESC, HNSC) and preprocessed data following Pathformer [46] (Details in Appendix B.2). Every patient (i.e., a hypergraph) has the same genes (i.e., nodes) and pathways (i.e., hyperedges), but the clinical data (i.e., gene features) are different. The data statistic of each cancer data is provided in Appendix B.1.

**Compared Methods.** We compared Natural-HNN with HNNs introduced in Section 2.3. Specifically, HGNN[20], HCHA [3], HNHN [12], UniGCNII [32], AllDeepSets [6], AllSetTransformer [6], HyperGAT [11], SHINE [48], ED-HNN [70], ED-HNNII [70] and a hypergraph disentangling method HSDN [31] are used as baselines. Implementation details of some baselines and their variants are described in Appendix C.1.

**Evaluation.** We randomly split the data into 50%/25%/25% for training/validation/test set. We measured average and standard deviation of the performances for 10 different data splits. The hyperparameter search space is provided in Appendix C.5.

### 5.2 Results for Cancer Subtype Classification (RQ1)

The cancer subtype classification task can be considered as a hypergraph classification task, since every patient (i.e., a hypergraph) has the same genes (i.e., nodes) and pathways (i.e., hyperedges). Specifically, we generated the representation of a hyperedge by simply concatenating representations of hyperedges in a hypergraph following Pathformer [46], due to the lack of an effective pooling method reflecting the hypergraph topology developed to date. Then, we applied one layer MLP as the classifier. We have the following observations in Table 1. **1)** Natural-HNN shows superior performance in most of the cancers with large performance gap compared with most of the models. Especially in the case of BRCA, we achieve approximately a 5% performance improvement compared to the second-best model. It can be concluded that incorporating the functional context (i.e., factors) of pathways has contributed to improved performance. **2)** Natural-HNN outperforms the hypergraph-structure-level disentanglement model, HSDN, with a significant performance gap. HSDN uses a factor similarity-based criterion to determine the relevance of factors. However, the superior performance of Natural-HNN validates that naturality-guided disentanglement is more effective at integrating the context behind group interactions.

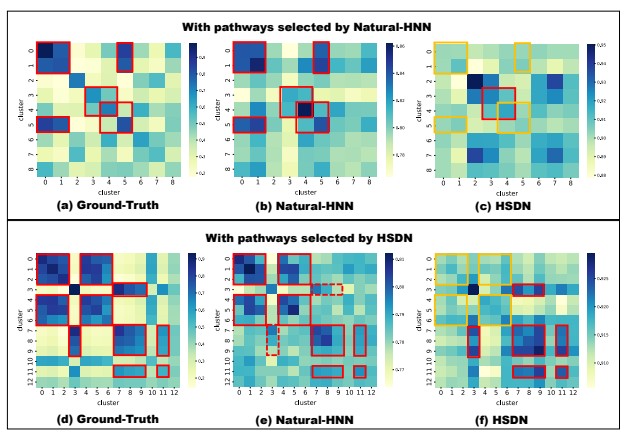

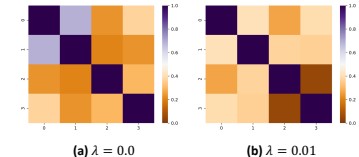

(a) $\lambda = 0.0$    (b) $\lambda = 0.01$

Figure 6: Pearson correlation between hyperedge factors.

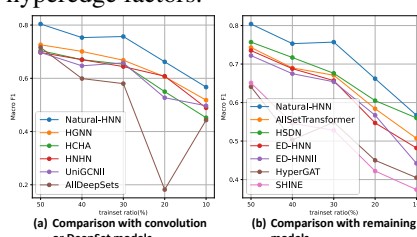

(a) Comparison with convolution or DeepSet models    (b) Comparison with remaining models

Figure 5: Captured interaction context. Captured patterns are shown in red boxes and not captured patterns are shown with orange boxes. Weakly captured cases are marked as dotted red block.

Figure 7: Marcro F1 scores with different training set ratio.

## 5.3 Capturing the Interaction Context of Hyperedges (RQ2)

To validate that Natural-HNN captured factors relevant to hyperedge disentanglement, we checked whether our model captures the functional semantics of genetic pathways. Because the models rely solely on cancer subtype labels during training[4], we expect the interaction contexts of informative hyperedges (such as cancer-related pathways) to be captured by the models, while non-informative hyperedges (such as pathways not relevant to cancer) are not. For this experiment, we first selected top-15 pathways[5] based on the SHAP value for each model (Natural-HNN in Figure 5 top and HSDN in Figure 5 bottom). Note that we rely on the SHAP value since information regarding which pathways are relevant to cancers is not given. Then, after clustering these 15 pathways with CliXO algorithm [34], we calculate the similarity between clusters based on the average similarity of pathways that belong to each cluster. Our goal is to check how well Natural-HNN preserves the functional semantic similarity between pathway clusters compared with the cluster similarity calculated with Lin's method [44] (BMA), which we consider as the ground-truth. For HSDN and Natural-HNN, cluster similarity is calculated based on the relevance score vector of each hyperedge $e_i$ across all factors, i.e., $\alpha_i = [\alpha_i^1, ..., \alpha_i^K]$, which can be calculated as $1/(1 + \|\alpha_i - \alpha_j\|_2)$. As the experiment setting is somewhat complicated, we described the detailed procedure in Appendix B.3.

The result on the BRCA datset is shown in Figure 5. The row and column of each heatmap is the index of the pathway clusters and color represents similarity between clusters. Figure 5 (a), (b) and (c) shows the measured similarity between clusters with pathways selected by Natural-HNN. Comparing (b) and (c) with (a), we observe that Natural-HNN preserves the functional similarity (red box) better than HSDN, which fails to do so (orange box). Moreover, Figure 5 (d), (e) and (f) shows the measured similarity between clusters with pathways selected by HSDN. An interesting observation is that even with the pathways that were informative to the HSDN, HSDN fails (orange box) to preserve the functional similarity between clusters while Natural-HNN could capture them. The results imply that the naturality condition in category theory is effective in capturing the interaction context of a hyperedge.

Finally, we checked whether each factor captures a different context by calculating Pearson correlation coefficients among hyperedges captured by each factor, following [85]. As shown in Figure 6 (b), factors tend to exhibit only weak correlations. Note that even when factors are completely disentangled, a small degree of correlation can naturally exist between factors, as described in [59]. We observe that the factor discrimination loss decreases correlation between factors when comparing Figures 6 (b) and (a).

---

[4]This means that models do not use external data related to pathway types or pre-trained models.

[5]Only a few pathways are related to each type of cancer. We can also observe this with the SHAP value distribution in Figure 15 of Appendix B.4.

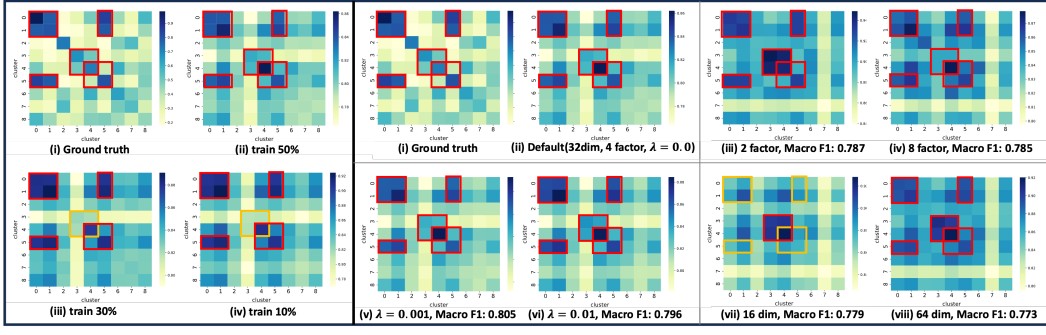

**(a) Training set ratio**  **(b) Hyperparameters**

Figure 8: Captured functional context with different (a) Training set ratio and (b) Hyperparameters. Patterns that are well-captured are shown in red and those that are not captured are shown in orange.

### 5.4 Generalizability and hyperparameter sensitivity of Natural-HNN (RQ3)

**Generalizability**. To validate the generalizability of Natural-HNN, we measured performance while gradually reducing the training set ratio from 50% to 10% in 10% decrements. Figure 7 shows the performance of our model (blue) and baselines. Figure 7 (a) shows a comparison with convolution-or DeepSet-based models. These baselines rely on a strong inductive bias that nodes contribute equally to hyperedges during the group interaction process. Models with such strong inductive biases typically exhibit strong generalizability. Observing the extent of performance degradation as the training ratio decreases, we can see that Natural-HNN also demonstrates good generalizability. Figure 7 (b) compares Natural-HNN with attention-based models, which are known for their strong expressivity. As shown in the figure, Natural-HNN consistently outperforms these models. This indicates that Natural-HNN possesses both strong generalizability and sufficient expressivity. Figure 8 (a) presents experimental results evaluating whether the functional context (i.e., factors) is well captured even as the training ratio decreases. As can be seen, a significant portion of the functional context is well captured despite the reduced training data, demonstrating that our proposed criterion effectively captures the factors.

**Hyperparameter Sensitivity**. We conducted experiments to evaluate the impact of hyperparameters, such as the number of factors, on Natural-HNN's ability to capture factors. Figure 8 (b) reveals the following insights: **1)** When the number of factors is 2 or 8, the overall similarity tends to be slightly higher than the ground truth; however, the core strong similarities are still well captured. **2)** Regardless of the value of the factor discrimination loss weight $\lambda$, the functional context (factors) is consistently well captured. **3)** When the dimensionality is too large, the core strong similarities are well captured, but the overall similarity tends to be slightly higher than the ground truth. Conversely, when the dimensionality is too small, some functional similarities are missed. These observations suggest that, except when the dimensionality is too small, Natural-HNN can generally capture the functional context well, regardless of hyperparameter settings.

**Additional Experiments.** In the Appendix, we provide ablation studies (Appendix E), time complexity analysis (Appendix F.1) and results on hypergraph benchmark datasets (Appendix D).

## 6 Conclusion

In this work, we propose a criterion for hyperedge disentanglement by discovering a characteristic called factor representation consistency. To uncover this characteristic, we analyzed the compositional structure in hypergraph message passing and focused on the naturality condition that is satisfied between entangled and disentangled representations. The characteristic derived from a hyperedge disentanglement model that structurally aligns with the underlying mechanism demonstrated effectiveness in capturing the functional context (i.e., factors) of genetic pathways (i.e., group interactions). Experiments showed that this simple criterion generalizes well and consistently captures factors regardless of hyperparameter choices.

## Acknowledgments and Disclosure of Funding

This work was supported by the Institute of Information & Communications Technology Planning & Evaluation(IITP) grant funded by the Korea government(MSIT) (RS-2025-02304967, AI Star Fellowship(KAIST)). Additionally, this work was supported by Institute of Information & communications Technology Planning & Evaluation (IITP) grant funded by the Korea government(MSIT) (No.2020-0-00004) Finally, this work was supported by National Research Foundation of Korea(NRF) funded by Ministry of Science and ICT (RS-2022-NR068758). The results shown here are in whole or part based upon data generated by the TCGA Research Network: https://www.cancer.gov/tcga.

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

# Appendix

# A  Category Theory

## A.1  Category Theory

Category theory [23, 39] is widely used to represent and analyze the structure or relation of a system. Instead of focusing on the details, category theory takes bird's eye view to see global structure and patterns. Recently, category theory is used to explain learning mechanism of machine learning methods [5, 40, 25, 22, 24, 9, 63, 10, 4, 82, 13, 14, 81]. In this paper, we only use simple, fundamental concepts of category theory: category, functor, natural transformation and product.

## A.2  Category

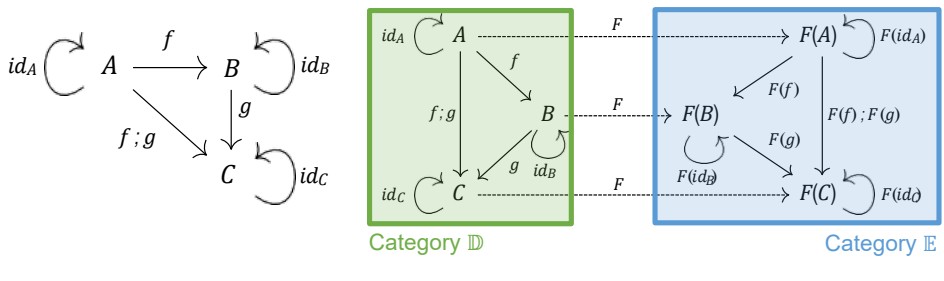

(a) Category                                        (b) Functor

Figure 9: Category and Functor

A category $\mathbb{C}$ is contains four components: collection of objects, morphisms, composition rule and identities.

- Collection of objects : $Ob(\mathbb{C})$ (ex : $\{A, B, C\}$ in Figure 9 (a))
- For every pair of objects $A, B \in Ob(\mathbb{C})$, there exists a set $Hom_{\mathbb{C}}(A, B)$. Element of the set is morphism and is denoted as: $f : A \to B$.
- For every three objects $A, B, C \in Ob(\mathbb{C})$, morphisms $f \in Hom_{\mathbb{C}}(A, B)$ (i.e. $f : A \to B$) and $g \in Hom_{\mathbb{C}}(B, C)$ (i.e. $g : B \to C$), **composition rule** holds : $f \,\overset{\circ}{,}\, g = g \circ f \in Hom_{\mathbb{C}}(A, C)^6$.
- For every object $A \in Ob(\mathbb{C})$, there exists an identity morphism $id_A \in Hom_{\mathbb{C}}(A, A)$ satisfying the following : $id_A \,\overset{\circ}{,}\, f = f = f \,\overset{\circ}{,}\, id_B$ for morphism $f : A \to B$.

Fig. 9 (a) shows an example of a category with three objects $(A, B, C)$. For each object, there is an identity morphism $(id_A, id_B, id_C)$. For every object pair, there is morphism $(f, g, f \,\overset{\circ}{,}\, g)$ with composition rules.

One of the most important categories is **Set**. In **Set**, the objects are sets and morphisms are functions mapping two sets. The composition rule is satisfied since a composition of two functions becomes a function. Another important category is category of relations, which is denoted as **Rel**. The objects of **Rel** are sets and relations $R \subseteq A \times B$ are morphisms between objects $A$ and $B$. Partially ordered set or poset can be considered as a category where objects are sets and morphisms are partial orders $\leqslant$. Since partial order is a kind of a relation, we can consider this category is a kind of **Rel**.

In Section 3, we analyzed hypergraph message passing framework, and found that, as nodes (considering node as set) are included in hyperedges, hypergraph message passing framework has poset structure with inclusion maps between them. We will define it **PISet**, a category for poset with inclusion morphisms (object is a set, morphisms are inclusions). Since inclusions are partial orders, which is also a relation, we can consider **PISet** as a kind of **Rel** category.

We can define our own category, similar to the one in a prior work [62], such that objects are vector representations and their (linear or non-linear) transformations are morphisms. We will call this a 'category of Deep Learning Representations' and denote **DLRep**.

---

[6]Two notations $f \,\overset{\circ}{,}\, g$ and $g \circ f$ have the same meaning : "applying $f$ first, and then applying $g$"

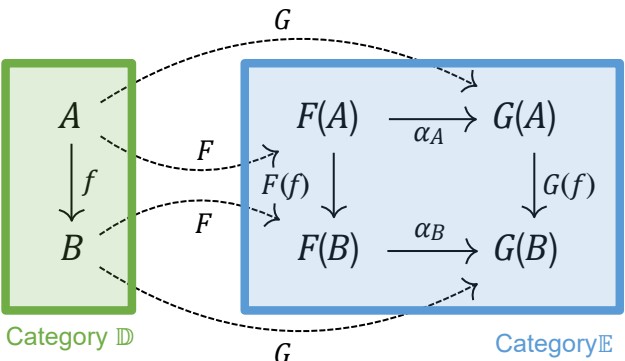

Figure 10: Natural transformation. Identity morphisms are omitted in the figure for simplicity.

## A.3 Functor

Functor is a structure preserving map between categories. Objects and morphisms in one category are mapped to objects and morphisms in different category, respectively. Figure 9 (b) shows an example of a functor mapping from category $\mathbb{D}$ to category $\mathbb{E}$. Each object in category $\mathbb{D}$ (i.e., $A, B, C$) is mapped to objects in category $\mathbb{E}$ (i.e., $F(A), F(B), F(C)$). The morphisms, including identity morphism, and their compositions in category $\mathbb{D}$ (i.e., $id_A, id_B, id_C, f, g, f \mathbin{\raise0.3ex\hbox{$\scriptstyle\circ$}} g$) are also mapped to morphisms in category $\mathbb{E}$ (i.e., $F(id_A), F(id_B), F(id_C), F(f), F(g), F(f) \mathbin{\raise0.3ex\hbox{$\scriptstyle\circ$}} F(g)$). In a metaphorical sense, functors serve as bridges that connect two distinct realms while maintaining an identical compositional structure[7].

One example can be a functor mapping from **Set** to **DLRep**. Each set (object) in **Set** is mapped to a vector representation (object) in **DLRep**. Functions (morphisms) in **Set** are mapped to transformations (morphism) between vector representations in **DLRep**. This functor is related to representation learning, since entities (i.e. concept or set) are mapped to their vector representations preserving their compositional structure (relation).

## A.4 Natural Transformation

Given two functors mapping from one category to another category, i.e., $F$ and $G : \mathbb{D} \to \mathbb{E}$, natural transformation is a way of relating these two functors using morphisms in target category $\mathbb{E}$. Specifically, for each object $A \in \mathbb{D}$, there exists a morphism $\alpha_A : F(A) \to G(A)$ in $\mathbb{E}$. The natural transformation must satisfy the following condition. For every morphism $f : A \to B$ in $\mathbb{D}$,

$$F(f) \mathbin{\raise0.3ex\hbox{$\scriptstyle\circ$}} \alpha_B = \alpha_A \mathbin{\raise0.3ex\hbox{$\scriptstyle\circ$}} G(f) \tag{2}$$

must hold. This condition is called the ***naturality condition***. Figure 10 shows an example of natural transformation. Functors $F$ and $G$ map objects and morphisms in category $\mathbb{D}$ to category $\mathbb{E}$. Natural transformation $\alpha : F \Rightarrow G$ maps $F(A)$ and $F(B)$ with $\alpha_A$ and maps $G(A)$ and $G(B)$ with $\alpha_B$. The objects and morphisms mapped by two functors as well as natural transformation $\alpha$ all belong to the category $\mathbb{E}$. Thus, natural transformation can be seen as a way of relating different views using morphisms in $\mathbb{E}$[8].

---

[7]The typical example of deep learning method using this concept is sheaf neural network [26], motivated from cellular sheaf [27]. There are also numerous studies in data science with a similar perspective [50, 69, 36].

[8]One typical example of deep learning method using this concept is Natural Graph Networks [10].

$$Y \xrightarrow{f_1} \quad \langle f_1, f_2 \rangle \quad \xrightarrow{f_2}$$

$$X_1 \xleftarrow{p_1} X_1 \times X_2 \xrightarrow{p_2} X_2$$

Figure 11: Product

$$Y$$
$$f_1 \quad [f_1, f_2] \quad f_2$$
$$X_1 \xrightarrow{i_1} X_1 \amalg X_2 \xleftarrow{i_2} X_2$$

Figure 12: Coproduct

$$X_1 \xleftarrow{p_1} X_1 \times X_2 \xrightarrow{p_2} X_2$$
$$\downarrow f_1 \qquad \downarrow f_1 \times f_2 \qquad \downarrow f_2$$
$$Y_1 \xleftarrow{q_1} Y_1 \times Y_2 \xrightarrow{q_2} Y_2$$

Figure 13: Product of morphisms.

## A.5 Product

**Product (Objects)** Let $\mathbb{C}$ be a category. For two objects $X_1, X_2 \in Ob(\mathbb{C})$, one can define product of two objects $X_1 \times X_2$ with morphisms $p_1 : X_1 \times X_2 \rightarrow X_1$ and $p_2 : X_1 \times X_2 \rightarrow X_2$ which are called **projections**. Then, the composition of objects in Figure 11 must be satisfied. Given object $Y \in Ob(\mathbb{C})$ with two morphisms $f_1 : Y \rightarrow X_1$ and $f_2 : Y \rightarrow X_2$, there exists a unique morphism called 'paring' [84] $\langle f_1, f_2 \rangle : Y \rightarrow X_1 \times X_2$ that satisfies the composition : $f_1 = \langle f_1, f_2 \rangle \, \mathring{,} \, p_1$ and $f_2 = \langle f_1, f_2 \rangle \, \mathring{,} \, p_2$.

### Coproduct (Objects)

A coproduct is the dual of a product, which can be obtained by reversing the direction of the arrows. Let $\mathbb{C}$ be a category. For two objects $X_1, X_2 \in Ob(\mathbb{C})$, one can define coproduct of two objects $X_1 \amalg X_2$ with morphisms $i_1 : X_1 \rightarrow X_1 \times X_2$ and $i_2 : X_2 \rightarrow X_1 \times X_2$ which are called **injections**. Then, the composition of objects in Figure 12 must be satisfied. Given object $Y \in Ob(\mathbb{C})$ with two morphisms $f_1 : X_1 \rightarrow Y$ and $f_2 : X_2 \rightarrow Y$, there exists a unique morphism $[f_1, f_2] : X_1 \amalg X_2 \rightarrow Y$ that satisfies the composition : $f_1 = i_1 \, \mathring{,} \, [f_1, f_2]$ and $f_2 = i_2 \, \mathring{,} \, [f_1, f_2]$.

### Product of Morphisms

Let $\mathbb{C}$ be a category. For objects $X_1, X_2, Y_1, Y_2 \in ob(\mathbb{C})$ and morphisms $f_1 : X_1 \rightarrow Y_1$ and $f_2 : X_2 \rightarrow Y_2$, we can define **product of morphisms** $f_1 \times f_2 : X_1 \times X_2 \rightarrow Y_1 \times Y_2 := \langle p_1 \, \mathring{,} \, f_1, p_2 \, \mathring{,} \, f_2 \rangle$ satisfying the compositional structure shown in Figure 13.

## A.6 Derivation of Figure 3 (b) from Figure 3 (a).

Since we are dealing with the commutative diagram between entangled and disentangled representations, we focus on the morphisms between $X^{en}, X^{dis}, H^{en}, H^{dis}$ in Figure 3 (a). Since the morphism (i.e., $f_c^{dis} \times f_d^{dis}$) in the disentangled representation is the product of factor-specific morphisms $f_c^{dis}$ and $f_d^{dis}$, we apply the diagram at Figure 13 to $f_c^{dis} \times f_d^{dis}$. Then we can get the morphisms between $H_c^{dis}, H^{dis}, H_d^{dis}, X_c^{dis}, X^{dis}, X_d^{dis}$ in the Figure 14 where $H^{dis} = H_c^{dis} \times H_d^{dis}$ and $X^{dis} = X_c^{dis} \times X_d^{dis}$. Note that morphisms between $H^{en}, H_c^{dis}, H^{dis}, H_d^{dis}$, are products shown in the Figure 11. If we extract components in Figure 14 that are related to factor 'c' and entangled representation, we have the diagram in Figure 3 (b).

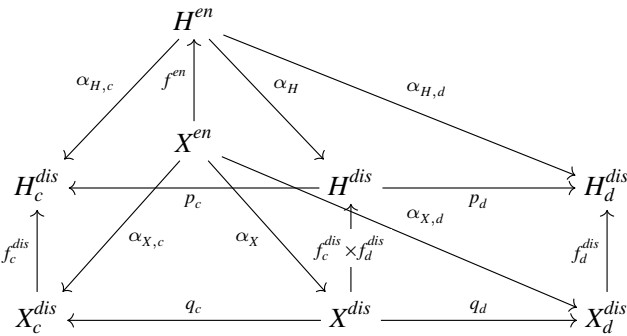

Figure 14: Derivation of Figure 3 (b) from Figure 3 (a).

# B    Dataset and Experiment Details

Note that KIPAN and NSCLC are known to be cancers where subtypes can be easily classified based on features alone [71, 56]. Because these datasets offer limited value for evaluating model performance, we excluded them in this study. The results of Natural-HNN and baselines for these datasets (i.e., KIPAN and NSCLC) are reported in [38].

## B.1    Statistics : Cancer Subtype Classification Dataset

The statistics of cancer datasets are shown in the Table 2. Note that every hypergraphs in all 6 cancers have 1497 pathways (hyperedges) and 11552 genes (nodes) with 9 feature dimension. The degree statistics of cancer dataset is shown in the Table 3. When converted to a graph with star-expansion, the graph contains 98013 edges. When converted to a graph with clique-expansion, the graph contains 10114890 edges. Thus, converting the hypergraph into a graph with clique-expansion requires large computation during message passing. The downloading and preprocessing details are provided in Appendix B.2.

Table 2: Statistics of 6 cancer datasets used for cancer subtype classification task.

| dataset | summary | class distribution(counts) |
|---------|---------|----------------------------|
| BRCA | 5 class, 769 hypergraphs | Normal-like 33, Her2 44, Basal-like 134, LumB 143, LumA 415 |
| STAD | 5 class, 341 hypergraphs | CIN 200, EBV 29, GS 46, MSI 59, HM-SNV 7 |
| SARC | 4 class, 257 hypergraphs | LMS 104, MFS/UPS 75, DDLPS 57, Other 21 |
| LGG | 2 class, 503 hypergraphs | G2 242, G3 261 |
| HNSC | 2 class, 507 hypergraphs | HPV- 411, HPV+ 96 |
| CESC | 2 class, 280 hypergraphs | AdenoCarcinoma 46, SquamousCarcinoma 234 |

Table 3: statistics of hypergraphs in cancer subtype classification task

|  | min | median | mean | max | std |
|--|-----|--------|------|-----|-----|
| node degree | 2 | 5 | 8.485 | 239 | 13.301 |
| hyperedge degree | 13 | 35 | 57 | 1371 | 84.720 |

## B.2    Preprocessing : Cancer Subtype Classification Dataset

The overall procedure was adopted from Pathformer [46]. However, statistics of the data can be slightly different due to the difference of time at which the data was downloaded.

### Creating Hypergraph

We downloaded pathways from several pathway databases including KEGG [33], PID [61], Reactome [8] and Biocarta.[55]. The pathways were selected based on their size and overlap ratio with other pathways. These two conditions must be considered as 1) extremely large pathways do not represent specific functions but rather general functions, 2) small pathways complicate interpretations 3) overlapping pathways cause redundancies. The more detailed explanations can be found in [58]. Pathways with too small or too big size or large overlaps are excluded. A specific threshold was chosen following the Pathformer.

### Generating Hypergraph Labels

For BRCA and STAD, we gathered cancer subtypes from TCGA [73] using TCGAbiolinks [7, 64, 53] R library. For the rest of 4 cancer datasets we downloaded cancer subtypes from Broad GDAC Firehose (https://gdac.broadinstitute.org/)[9].

---

[9]Pathformer used labels from pan-cancer atlas study [60] for HNSC, CESC and SARC. However, we decided to use the one in Broad GDAC Firehose since it was easier to process the same data

**Generating Node Features**

We gathered mRNA/miRNA expression, DNA methylation[10], DNA copy number variation (CNV)[11] using TCGAbiolinks. Gene lengths were acquired from biomaRt R package [17, 16]. The procedure of processing each data with Gistic2 [52], normalization by TPM are adopted from Pathformer. At the end of the processing step, we calculate statistics (mean, min, max, count) of modalities as values for each feature dimension.

## B.3 Experiment Details of Capturing Context Types

To check whether HNNs could capture functional semantics of pathways (i.e, interaction context of hyperedges), we need functional context annotations for each hyperedge. However, there is no data that annotates the functional semantics of genetic pathways. Instead, to assign function-related hyperedge types or labels, we clustered pathways based on the functional similarity between pathways, which can be calculated with computational biology method.

Now that we have obtained the hyperedge types, one might think we can simply check whether there is a one-to-one correspondence between hyperedge types and factors. However, there is another issue: hyperedge types themselves can be similar to each other. In other words, due to functional correlations between hyperedge types, a single factor may appear not in just one hyperedge type but across multiple hyperedge types. Therefore, examining the relationship between factors and hyperedge types alone makes it difficult to determine whether disentanglement has captured the functional context. Instead, we can indirectly verify that factors are related to the functional context by checking whether the functional similarity between hyperedge types aligns with the functional similarity inferred from the model's factor relevance. Thus, we evaluated whether the model effectively captured the functional context by comparing the ground truth functional similarity between hyperedge types (i.e., clusters) with the similarity inferred from the model. If the functional similarity predicted by the model shows some correlation with the functional similarity defined as ground truth, we can say that the model has captured the functional context. We do not directly compare the exact values of prediction and the ground truth since the way of calculating the value is different in prediction (calculation based on relevance scores $\alpha_{e_i}^k$) and ground truth (algorithm used in computational biology). Therefore, instead of comparing exact values, we assessed it based on the similarity of patterns observable in a heatmap, as shown in Figure 5.

In summary, our experiment involves selecting pathways to be analyzed, collecting function-related information for each pathway, measuring the functional similarity between pathways based on the collected information, and performing clustering based on this similarity. Afterward, we compute the similarity between clusters to derive the ground truth similarity, which is then compared with the model's predictions. Thus, in order to perform the experiment, we need to consider the followings: **1)** Which pathways need to be analyzed? **2)** How to get ground truth pathway functions? (i.e. How to get function related information?) **3)** How to calculate ground truth functional similarity between pathways **4)** How to cluster functionally similar pathways in a reliable manner **5)** How to measure ground truth cluster similarity and how to predict cluster similarity with model outputs.

**Which pathways need to be analyzed?** There are two reasons behind selecting pathways : 1) Since CliXO algorithm (Appendix B.6) used for clustering pathways takes a lot of time, the number of pathways to be analyzed must be reduced. 2) The ground truth functional similarity (Appendix B.5) contains vast biological context derived from biological domain knowledge or researches, which might not be present in our dataset. Since our dataset contains only cancer-specific information, there is no way to capture non-existing context (contexts that are not related to cancer) without external supervision. Thus direct comparison between the ground truth and our result is impossible. The most ideal way for fair comparison would be selecting the ground truth that is only relevant to our dataset or task. However, it is impossible since there are no databases with annotated context (cancer or environment) specific pathway functionalities. An alternative way was selecting the pathways that were informative or important in the decision of the model. If a model can correctly capture functional context of pathways, since pathway functions are highly related to the cancers [74, 65], informative pathways (for the model prediction) are the pathways that contain cancer-specific contexts. Since we only need to check whether functional context are correctly captured under the cancer specific

---

[10]but we do not use promoter methylation
[11]but we do not use gene level CNV

circumstances or condition, by selecting those pathways, we can compare functional similarities that are specific to our data or cancer[12]. The details for selecting pathways are described in Appendix B.4.

**How to get ground truth pathway functions.** Since there is no database that annotates functional similarity scores between pathways, we rely on methods used in computational biology. Hence, we need to get pathway function information. Similarity calculations and clusterings are based on the annotation of pathway functions. The details are described in Appendix B.5.

**How to calculate ground truth functional similarity between pathways.** Based on the functions of pathways, pathway functional similarity can be calculated. The calculated similarity will be used in clustering and generating ground truth functional similarity between clusters. The details are dealt in Appendix B.5.

**How to cluster functionally similar pathways in a reliable manner.** With functional similarity between pathways, we can cluster functionally similar pathways with CliXO algorithm. The details and example results are shown in Appendix B.6.

**How to measure ground truth cluster similarity and how to predict cluster similarity with model outputs.** Finally, we need to devise a way to measure the similarity between clusters based on the model outputs. Also, we need to measure ground truth functional similarity between clusters. The details are described in Appendix B.7.

In summary, the procedure of experiments can be described as follows. First, we get functional annotation of pathways (hyperedges). Second, we calculate functional similarity between pathways based on annotations. Third, we select pathways to be analyzed based on the model output. Fourth, we cluster the selected pathways with pathway similarity. Finally, we calculate the predicted functional similarity between clusters from model prediction and compare that with the ground truth cluster similarity.

## B.4    Selecting Pathways with SHAP values

To select pathways that were the most informative for prediction, we provide the final representation of pathways generated by a model, 1 layer classifier (MLP) that predicts labels from final representation as well as labels to the DeepExplainer to get SHAP values. Then we select top-k pathways based on the SHAP value. Note that only small number of pathways are relevant to the task as shown in Figure 15. This is due to the fact that not all pathways are related to very specific type of cancer. Although Natural-HNN and HSDN both use the same number of pathways (top-k), the pathways selected by each model can be different. This also leads to different number of clusters in Figure 5 and 18.

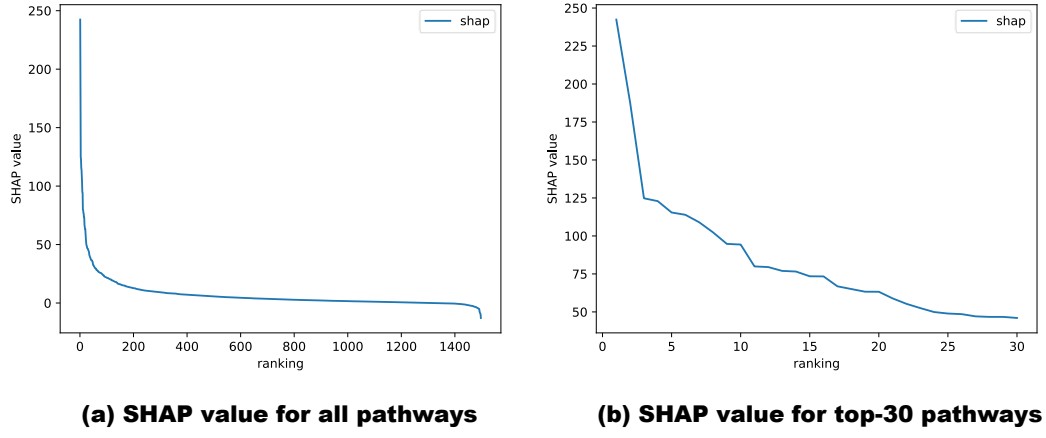

(a) SHAP value for all pathways          (b) SHAP value for top-30 pathways

Figure 15: SHAP value distribution of Natural-HNN on BRCA dataset. We sorted pathways with SHAP value. X axis represents ranking of pathways and Y axis represents SHAP value for pathways with corresponding ranking.

---

[12]On the other hand, if the model could not correctly capture pathway functionalities, cancer irrelevant pathways will be selected and will have different result from the ground truth in section 5.3

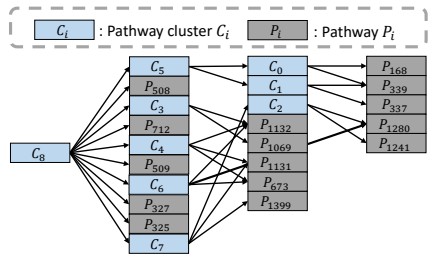

**(a) Clustering result for (SHAP value) top 15 pathways of Natural-HNN @ BRCA**

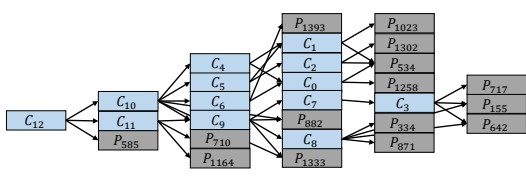

**(b) Clustering result for (SHAP value) top 15 pathways of HSDN @ BRCA**

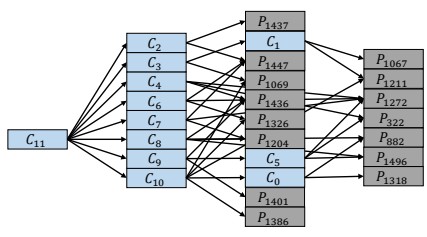

**(c) Clustering result for (SHAP value) top 15 pathways of Natural-HNN @ CESC**

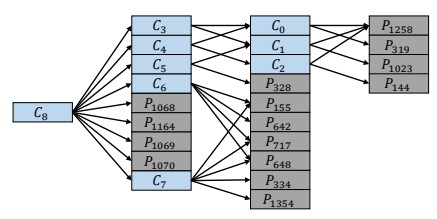

**(d) Clustering result for (SHAP value) top 15 pathways of HSDN @ CESC**

Figure 16: The result of applying CliXO algorithm to top-15 pathways of Natural-HNN and HSDN on BRCA and CESC. The pathway number denotes the index of pathway in our dataset (hyperedge index).

## B.5 Calculating Functional Similarity between Pathways

This process consists of two steps: 1) assigning pathway level function to pathways and 2) calculating functional semantic similarities between pathways. For both two steps, we adopted the most frequently used and verified methods through several studies. For the assignment of pathway functions, we use GO enrichment analysis. Gene ontology (GO) [2, 1] is a functional annotation of genes that has a hierarchical structure. Note that, however, the hierarchical structure of functional annotations is close to a directed acyclic graph (DAG) rather than a tree-like hierarchical structure. As an example, we can see DAG structure in the result of CliXO algorithm in the Figure 16. We can computationally annotate pathway functions with GO terms using GO enrichment analysis. We use 'enrichGO' function provided by R package clusterProfiler [80], with pvalue of 0.01 followig the paper [65]. Then we selected the most specific GO terms with set cover algorithm proposed in [65] to assign pathways precise representation of their functions.

The next step is calculating functional semantic similarities between pathways. We used Lin's method [44] with best matching average (BMA) as the combination was proven to perform well with CliXO and was proven to be robust in incomplete annotation cases in [45]. We used mgoSim function in R package GOSemSim [79, 78] for the calculation of Lin's method.

## B.6 Assigning Pathway Type with CliXO

To cluster functionally similar pathways, we adopted CliXO [34]. It was originally designed to cluster gene function annotations (GO) and has been used in multiple biological studies[35, 57]. However, it can also be effectively applied to higher functional semantics such as pathways as in [86]. We used official implementation of CliXO 1.0 for our research. We used the following 4 values as hyperparameter of CliXO : a = 0.1, b = 0.6, m = 0.005, s = 0.2.

Since CliXO can cluster functionally similar pathways, we can assign interaction types to pathways by assigning them to the cluster. Figure 16 shows the result of applying CliXO for top-15 pathways selected by Natural-HNN or HSDN for BRCA as well as CESC. Unlike other hierarchical clustering based methods, CliXO created clusters having DAG structure. Considering that GO also has DAG structure, CliXO can be seen as a natural way of reflecting complex structure or relations in biology.

## B.7 Calculating Functional Similarity between clusters

**Ground Truth** Given a pair of clusters, calculating functional similarity between them is simple. We average the similarity of all possible pathway pairs belonging to different clusters to get functional similarity between clusters.

**Model's prediction** If a model correctly captures functional context of pathways, then the relevance scores ($\alpha_i^k$) of two similar pathways must be similar for all factors. Thus we define the similarity between pathways as $\frac{1}{1+\|\alpha_i-\alpha_j\|_2}$, where $\alpha_i = [\alpha_i^1, ..., \alpha_i^K]$ is a factor vector of pathway (hyperedge) $e_i$. The cluster similarity can be calculated in the same way as in the ground truth case. We average the similarity of all possible pathway pairs belonging to different clusters to get functional similarity between clusters.

# C Implementation Details

In Appendix C.1, we describe some implementation details of baselines and their variants, which can be different from official implementations. From Appendix C.2 to C.5, we describe implementation details for the components of Natural-HNN.

## C.1 Baselines and their variants

We implemented HyperGAT based on the paper as its official implementation is different from what is explained in the paper. Moreover, as the original version of SHINE and HyperGAT do not involve multihead attention, we implement it for fair comparisons. For SHINE, we also implemented two versions, one without using $\mathcal{L}_{reg}$ and the other with $\mathcal{L}_{reg}$ which is a loss introduced by the paper for the purpose of making node representations to be similar if the nodes are included in the same hyperedge. However, we did not use the version with $\mathcal{L}_{reg}$ in cancer subtype classification task since the loss converts a hypergraph to a graph using clique expansion, which causes tremendous computational cost.

## C.2 Factor Discrimination Loss

We defined a factor discrimination loss $\mathcal{L}_{dis}$ similar to the one used in [85]. In order to promote factors to contain different information, we use a factor classifier implemented with one layer MLP. Each factor representation of every hyperedge will be given as input to the factor classifier. The classifier needs to identify to which factor the factor representation belongs. If the classifier can correctly identify the factor with factor representation, i.e. if factor representations of two different factors of a hyperedge are distinguishable, it is highly likely that factors contain different information.

Specifically, we can calculate the loss by creating pseudo labels. For each factor representation of each hyperedge ($h_{e_i}^k$), we assign a pseudo label $Y_{e_i}^k = k$. Then the loss can be defined as follows:

$$\mathcal{L}_{dis} = - \sum_{e_i \in \mathcal{E}} \sum_{k=1}^{K} \sum_{c=1}^{K} \mathbf{1}(Y_{e_i}^k = c) log(softmax(MLP(h_{e_i}^k))) \tag{3}$$

This loss is applied to each layer of Natural-HNN. As described in Section 4.4, the final loss would be $\mathcal{L} = \mathcal{L}_{task} + \lambda \mathcal{L}_{dis}$. As mentioned before, $\mathcal{L}_{dis}$ is an optional part of our model. The hyperparameter search space for $\lambda$ is provided in Appendix C.5

## C.3 Loss used for training $\mathcal{L}_{task}$

After the final message passing layer of Natural-HNN, we get the final node embeddings $z_{v_i}$. The classifier of Natural-HNN will predict labels $p_{v_i} \in \mathbb{R}^C$ where $C$ denotes the number of classes. In other words, $p_{v_i,c}$ denotes the probability that node $v_i$ has class $c$ as answer. If we denote $l_{v_i}$ as the label (one-hot vector) for node $v_i$, the task loss can be calculated with cross-entropy loss.

$$\mathcal{L}_{task} = - \sum_{i=1}^{|V|} \sum_{c=1}^{C} l_{v_i,c} \log(p_{v_i,c}) \tag{4}$$

Note that, we use hyperedge embedding of the final layer instead of node embeddings for cancer subtype classification task.

## C.4 Factor Encoder

In Section 4, we explained that we use $K$ number of MLPs to get $K$ factor representations. The resulting factor representation is a vector with size $d/K$ when desired output representation size of a layer is given as $d$. When implementing the factor encoder as a code, we use single MLP that outputs vector with size $d$. Note that applying $K$ different MLPs (with output vector size $d/K$) is the same as applying one MLP (with output vector size $d$) and chunking the vector to smaller ones with size $d/K$. (i.e. First $d/K$ values corresponds to the $1^{st}$ factor representation, and following $d/K$ values

Table 4: Hyperparameter search space in standard benchmark dataset. † : MLP layers used in AllDeepSets, AllSetTransforer, ED-HNN, ED-HNNII

| models | ♯ cl | classifier dim | head (factor) | ♯ MLP layer [†] | $\lambda$ for $\mathcal{L}_{dis}$ | ♯ Total |
|---|---|---|---|---|---|---|
| HGNN | 1 | - | 1 | - | - | 32 |
| HCHA | 1 | - | 1 | - | - | 32 |
| HNHN | 1 | - | 1 | - | - | 32 |
| UniGCNII | 1 | - | 1 | - | - | 32 |
| AllDeepSets | 1,2 | 64,128,256,512 | 1 | 1,2 | - | 320 |
| AllSetTransforer | 1,2 | 64,128,256,512 | 1,2,4,8 | 1,2 | - | 1280 |
| HyperGAT | 1 | - | 1,2,4,8 | - | - | 128 |
| SHINE | 1 | - | 1,2,4,8 | - | - | 128 |
| HSDN | 1 | - | 1,2,4,8 | - | 0.0001, 0.0005, 0.001, 0.005, 0.01, 0.05, 0.1 | 896 |
| ED-HNN | 1,2 | 64,128,256,512 | 1 | $[0,1,2] \times [1,2] \times [0,1,2]$ | - | 2880 |
| ED-HNNII | 1,2 | 64,128,256,512 | 1 | $[0,1,2] \times [1,2] \times [0,1,2]$ | - | 2880 |
| Natural-HNN | 1 | - | 2,4,8 | 1 | - | 96 |
| Natural-HNN+$\mathcal{L}_{dis}$ | 1 | - | 2,4,8 | 1 | 0.0001, 0.0005, 0.001, 0.005, 0.01, 0.05, 0.1 | 672 |

Table 5: Hyperparameter search space in cancer subtype classification task. † : MLP layers used in AllDeepSets, AllSetTransforer, ED-HNN, ED-HNNII

| models | head (factor) | ♯ MLP layer [†] | $\lambda$ for $\mathcal{L}_{dis}$ | ♯ Total |
|---|---|---|---|---|
| HGNN | 1 | - | - | 24 |
| HCHA | 1 | - | - | 24 |
| HNHN | 1 | - | - | 24 |
| UniGCNII | 1 | - | - | 24 |
| AllDeepSets | 1 | 1,2 | - | 48 |
| AllSetTransformer | 1,2,4,8 | 1,2 | - | 192 |
| HyperGAT | 1,2,4,8 | - | - | 96 |
| SHINE | 1,2,4,8 | - | - | 96 |
| HSDN | 1,2,4,8 | - | 0.0001, 0.0005, 0.001, 0.005, 0.01, 0.05, 0.1 | 672 |
| ED-HNN | 1 | $[0,1] \times [1] \times [0,1]$ | - | 96 |
| ED-HNNII | 1 | $[0,1] \times [1] \times [0,1]$ | - | 96 |
| Natural-HNN | 2,4,8 | - | - | 72 |

corresponds to the $2^{nd}$ factor representation and so on.) The nonlinear activation function we used for factor encoder is hyperbolic tangent (tanh).

## C.5 Hyperparameter search space

We report the hyperparameter search space of each model in standard benchmark dataset as well as cancer subtype classification task. We used Adam optimizer for Natural-HNN. For the baselines, we closely followed optimizers or schedulers they used in their paper. Table 4 and Table 5 shows the hyperparameter search space in the standard benchmark dataset and cancer subtype datasets respectively. '♯ Total' denotes the number of all possible hyperparameter combinations that each model needs to search. 'cl' denotes the number of classifier layers. When the number of classifiers is larger than 1, those models have an additional hyperparameter that decides the hidden dimension of the classifier. ♯ MLP layer denotes the number of layers in MLP that was used in AllDeepSets, AllSetTransformer, ED-HNN, ED-HNNII. In the case of ED-HNN and ED-HNNII, there were three types of MLPs and each MLP could have different number of layers. $\lambda$ for $\mathcal{L}_{dis}$ is hyperparameter that changes the reflection ratio of the factor discrimination loss.

For standard hypergraph benchmark datasets, we used [64, 128, 256, 512] as hidden dimension and [0.1, 0.01, 0.001, 0.0001] as learing rate. For weight decay, we used [0, 1e-5]. We fixed the number of layers to 2, except for HSDN, because HSDN uses only a single layer. Generally, we used 0.5 as dropout. (If the paper of a model specified dropout to a specific value, we used the value following the paper.) As we can see, our model generally has a small hyperparameter search space comparable to GAT (when not using $\mathcal{L}_{dis}$). Although ED-HNN and ED-HNNII had good performance on standard hypergraph benchmark datasets, they had to rely on very large hyperparameter search space.

For cancer subtype classification tasks, we used [16, 32, 64] as the hidden dimension and [0.1, 0.01, 0.001, 0.0001] as learning rate. For weight decay, we used [0, 1e-5]. We fixed the number of layers to 2, except for HSDN, because HSDN uses only a single layer. During training, we set 50 as the batch size. Generally, we used 0.5 as dropout. (If the paper of a model specified dropout to a specific value, we used the value following the paper.) Since we fixed the number of classifiers to 1, the hyperparameter search space of some models are largely reduced when compared to the node

classification task. For ED-HNN and ED-HNNII, we reduced the search space of the number of MLPs since it took too much time to get the results.

## C.6 Environment for experiment

We used 48GB NVIDIA RTX A6000 GPU. We created a anaconda environment with python 3.7.16, pytorch 1.11.0 and pytorch geometric with version 2.0.4. Details can also be found at https://github.com/Yoonho-Lee-AI4Science/Natural-HNN.

Table 6: Dataset statistics of standard hypergraph benchmark dataset

| | Cora | Citeseer | Pubmed | Cora-CA | DBLP-CA | NTU2012 | ModelNet40 | 20Newsgroups |
|---|---|---|---|---|---|---|---|---|
| # nodes | 2708 | 3312 | 19717 | 2708 | 41302 | 2012 | 12311 | 16242 |
| # edge | 1579 | 1079 | 7963 | 1072 | 22363 | 2012 | 12311 | 16242 |
| # feature | 1433 | 3703 | 500 | 1433 | 1425 | 100 | 100 | 100 |
| # classes | 7 | 6 | 3 | 7 | 6 | 67 | 40 | 4 |
| avg. $|e|$ | 3.03 | 3.200 | 4.349 | 4.277 | 4.452 | 5 | 5 | 654.51 |
| CE Homophily | 0.897 | 0.893 | 0.952 | 0.803 | 0.869 | 0.753 | 0.853 | 0.461 |

Table 7: Model performance on standard hypergraph benchmark datasets (Accuracy). The last row is the result with extreme hyperparameter search space that includes hyperparmeter searching for dropout and interpolation ratio $\beta$ (introduced in Section 4.3). Top three models (excluding the last row) are colored by **First**, **Second**, Third. † : the variant of the model using multihead attention. ⋆ : the variant of the model using $\mathcal{L}_{reg}$ defined in SHINE[48].

| Method | Cora | Citeseer | Pubmed | Cora-CA | DBLP-CA | NTU2012 | ModelNet40 | 20Newsgroups |
|---|---|---|---|---|---|---|---|---|
| HGNN | 79.453 ± 1.003 | 73.092 ± 1.582 | 87.336 ± 0.443 | 83.383 ± 1.028 | 91.410 ± 0.365 | 88.350 ± 1.082 | 95.567 ± 0.411 | 81.246 ± 0.435 |
| HCHA | 79.276 ± 1.158 | 73.693 ± 1.687 | 87.230 ± 0.511 | 83.191 ± 0.868 | 91.358 ± 0.374 | 88.270 ± 1.304 | 94.703 ± 0.283 | 81.189 ± 0.397 |
| HNHN | 76.765 ± 1.560 | 72.524 ± 1.570 | 87.237 ± 0.523 | 77.480 ± 0.932 | 86.927 ± 0.346 | 88.489 ± 0.878 | 97.811 ± 0.231 | 81.059 ± 0.485 |
| UniGCNII | 79.498 ± 1.508 | 73.514 ± 2.107 | 88.124 ± 0.376 | 83.840 ± 0.693 | 91.728 ± 0.225 | 89.245 ± 0.882 | 97.243 ± 0.334 | 81.687 ± 0.452 |
| AllDeepSets | 79.306 ± 1.627 | 72.959 ± 1.795 | 89.418 ± 0.360 | 84.594 ± 0.793 | 91.594 ± 0.308 | 88.847 ± 0.984 | 97.532 ± 0.185 | 81.721 ± 0.653 |
| AllSetTransformer | 79.749 ± 1.620 | 73.140 ± 1.804 | 88.667 ± 0.388 | 84.786 ± 0.690 | 91.593 ± 0.309 | 89.404 ± 1.074 | 98.217 ± 0.138 | 81.783 ± 0.569 |
| HyperGAT | 55.908 ± 4.128 | 41.751 ± 1.814 | 48.191 ± 0.443 | 73.560 ± 1.829 | 90.292 ± 0.468 | 83.857 ± 1.490 | 92.465 ± 0.387 | 80.997 ± 0.390 |
| HyperGAT† | 58.183 ± 2.079 | 42.246 ± 1.874 | 48.389 ± 0.426 | 73.752 ± 1.508 | 90.394 ± 0.362 | 85.467 ± 1.876 | 92.481 ± 0.463 | 81.083 ± 0.374 |
| SHINE | 57.755 ± 3.198 | 41.413 ± 0.680 | 48.576 ± 0.455 | 75.037 ± 1.912 | 90.759 ± 0.292 | 87.256 ± 1.393 | 93.803 ± 0.395 | 81.061 ± 0.632 |
| SHINE† | 56.307 ± 4.452 | 41.763 ± 0.693 | 48.576 ± 0.433 | 75.613 ± 1.508 | 90.697 ± 0.329 | 87.157 ± 1.426 | 93.878 ± 0.332 | 81.239 ± 0.459 |
| SHINE⋆ | 58.818 ± 1.591 | 41.413 ± 1.563 | 46.682 ± 1.177 | 74.623 ± 1.444 | 61.507 ± 12.169 | 81.451 ± 2.399 | 89.406 ± 0.775 | 61.492 ± 12.666 |
| SHINE†⋆ | 58.065 ± 1.616 | 41.123 ± 1.707 | 43.619 ± 1.402 | 73.087 ± 1.077 | 36.215 ± 17.676 | 70.835 ± 23.388 | 75.956 ± 23.688 | 56.452 ± 13.043 |
| HSDN | 76.632 ± 1.509 | 71.824 ± 1.779 | 87.193 ± 0.323 | 81.595 ± 1.011 | 90.229 ± 0.242 | 89.722 ± 1.196 | 83.439 ± 1.204 | 81.372 ± 0.435 |
| ED-HNN | 80.635 ± 1.670 | 73.696 ± 1.992 | 88.911 ± 0.410 | 85.480 ± 0.828 | 92.151 ± 0.291 | 87.594 ± 0.811 | 97.999 ± 0.199 | 81.608 ± 0.695 |
| ED-HNNII | 78.951 ± 1.445 | 72.524 ± 1.682 | 79.355 ± 0.953 | 83.693 ± 0.839 | 91.702 ± 0.325 | 86.223 ± 0.958 | 95.749 ± 0.335 | 80.150 ± 0.753 |
| Natural-HNN (ours) | 80.709 ± 1.635 | 73.285 ± 1.742 | 87.136 ± 0.450 | 84.993 ± 0.491 | 90.961 ± 0.137 | 89.900 ± 1.017 | 98.558 ± 0.295 | 81.734 ± 0.745 |
| Natural-HNN (ours + $\mathcal{L}_{dis}$) | 80.739 ± 1.570 | 73.551 ± 1.964 | 88.475 ± 0.466 | 85.081 ± 0.583 | 91.032 ± 0.179 | 90.060 ± 1.565 | 98.584 ± 0.254 | 81.827 ± 0.695 |
| Natural-HNN (ours, extreme) | 81.300 ± 1.323 | 74.058 ± 1.335 | 88.746 ± 0.511 | 85.583 ± 0.774 | 91.910 ± 0.192 | 90.417 ± 0.919 | 98.629 ± 0.229 | 82.083 ± 0.742 |

# D    Standard Hypergraph Benchmark dataset

We performed experiments with standard hypergraph benchmark dataset to check whether Natural-HNN can be applied to the **datasets that are not verified to have multiple factors behind group interactions. Considering how hyperedges were created for benchmark datasets, it is not likely that those datasets contain meaningful or task related interaction contexts.** In co-citation and co-authorship networks, for example, hyperedges are created by simply connecting all documents cited by a paper or written by an author. Citations between a pair of papers might have context that is related to a reason for citation, however, it is hard to expect that a group of documents (papers) cited by a paper creates a special meaning or have a special context. Even if we assume that hyperedges in co-citation networks contain interaction context, it is still not clear how these interaction contexts are related to the labels of nodes. It is also hard to expect interaction context in co-authorship networks for a similar reason. Thus, **the benchmark dataset experiment will verify whether Natural-HNN can be applied to the datasets where the existence of factors behind group interactions is not known.**

For the node classification task with standard hypergraph benchmark datasets, we randomly split the data into 50%/25%/25% for training/validation/test set. We measured average and standard deviation of the performances for 10 different data splits. The hyperparameter search space is provided in Appendix C.5.

## D.1    Statistics : Standard Hypergraph Benchmark Dataset

Cocitaion networks and coauthor networks are adopted from [76]. The node features are bag-of-words representation of each documents. NTU2012 and ModelNet40 dataset is computer vision and graphics datasets where features are generated by applying GVCNN[19] and MVCNN[66]. Node feature of 20Newsgroups are generated by TF-IDF representations of news. The statistics of standard benchmark dataset is given in Table 6. Homophily ratio was calculated after converting hypergraph into a graph with clique expansion (CE)[67] following the method described in the other work [70].

Table 8: Model performance on standard hypergraph benchmark datasets (Accuracy) trained with only 5% of data

| Method | Cora | Citeseer | Pubmed | Cora-CA | DBLP-CA | NTU2012 | ModelNet40 | 20Newsgroups |
|---|---|---|---|---|---|---|---|---|
| HGNN | 66.773 ± 2.806 | 61.445 ± 2.465 | 81.161 ± 0.531 | 71.548 ± 2.652 | 89.689 ± 0.384 | 58.884 ± 5.045 | 94.795 ± 0.381 | 79.690 ± 0.675 |
| HCHA | 67.403 ± 2.865 | 61.600 ± 2.279 | 81.135 ± 0.549 | 71.379 ± 2.465 | 89.689 ± 0.274 | 59.032 ± 5.083 | 93.939 ± 0.448 | 79.596 ± 0.652 |
| HNHN | 58.272 ± 1.970 | 58.473 ± 5.296 | 79.793 ± 0.804 | 58.831 ± 2.399 | 82.855 ± 0.499 | 58.737 ± 5.344 | 96.845 ± 0.382 | 78.456 ± 0.602 |
| UniGCNII | 68.212 ± 2.559 | 63.600 ± 1.203 | 83.024 ± 0.820 | 70.799 ± 2.606 | 88.751 ± 0.281 | 60.255 ± 5.022 | 96.584 ± 0.248 | 79.061 ± 0.506 |
| AllDeepSets | 65.694 ± 2.306 | 61.388 ± 4.012 | 84.485 ± 0.647 | 71.319 ± 2.964 | 59.689 ± 0.296 | 59.892 ± 4.833 | 96.055 ± 0.286 | 78.868 ± 0.534 |
| AllSetTransformer | 65.914 ± 2.155 | 62.506 ± 1.720 | 82.942 ± 0.491 | 71.249 ± 2.796 | 89.665 ± 0.216 | 60.444 ± 5.204 | 96.608 ± 0.291 | 79.409 ± 0.590 |
| HSDN | 58.332 ± 2.882 | 57.812 ± 1.808 | 80.195 ± 0.45 | 64.845 ± 4.025 | 87.636 ± 0.243 | 51.949 ± 17.016 | 97.159 ± 0.179 | 79.406 ± 0.594 |
| ED-HNN | 66.433 ± 2.824 | 61.759 ± 2.296 | 82.348 ± 0.559 | 69.809 ± 2.569 | 90.039 ± 0.342 | 57.984 ± 6.477 | 96.698 ± 0.265 | 78.386 ± 0.542 |
| Natural-HNN (ours) | 67.343 ± 1.837 | 62.620 ± 2.277 | 82.393 ± 0.467 | 70.809 ± 2.789 | 88.700 ± 0.251 | 60.511 ± 5.338 | 98.031 ± 0.196 | 79.329 ± 0.666 |
| Natural-HNN (ours + $\mathcal{L}_{dis}$) | 67.393 ± 1.938 | 62.694 ± 2.218 | 82.838 ± 0.609 | 70.909 ± 3.439 | 88.906 ± 0.204 | 61.384 ± 4.570 | 98.141 ± 0.116 | 79.431 ± 0.552 |

## D.2   Node Classification on Benchmark Datasets

Table 7 summarizes the node classification performance in standard hypergraph benchmark datasets. We have the following observations: **1)** Our model generally performs well on various datasets by taking the first or second place in terms of accuracy. In the case of Citeseer and Cora-CA, the performance of our model is comparable to the best performing model. The results indicate that our model can be applied to various circumstances, even when the context variety of hyperedges is not guaranteed. **2)** Attention-based models (i.e., AllSetTransformer, SHINE, and HyperGAT) and disentangle-based model (i.e., HSDN) generally perform similar to or worse than convolution-based models (i.e., HGNN, HCHA, HNHN, UniGCNII) and AllDeepSets (which also does not have heads or factors) on Citeseer, Pubmed and DBLP-CA. Through the results, we can guess that those datasets do not contain various interaction contexts that is helpful for the model performance. This can also be a reason why our model does not perform well on those datasets as much as on other datasets.

We consider a model that achieves sufficiently good performance without relying excessively on hyperparameter tuning to be reliable. However, there has been an increasing number of papers, such as Sheaf Hypergraph Networks [18], PhenomNN [72], and ED-HNN [70], that report performance obtained through an extreme level of hyperparameter tuning. Therefore, to enable a fair comparison with these works, we also included dropout and the interpolation ratio $\beta$ (introduced in Section 4.3) in the hyperparameter tuning and conducted additional experiments. For both dropout and the interpolation ratio $\beta$, we set the hyperparameter search space from 0.1 to 0.9 with an interval of 0.1. The results are reported in the last row of Table 7. Comparing the results of Natural-HNN with the official performance results of the papers mentioned earlier that rely on extreme hyperparameter tuning, we can see that Natural-HNN outperforms them despite having a much simpler model architecture.

## D.3   Training with only 5% of data

To check the generalization power of our model, we performed an experiment of training with only 5% of data. Following the split ratio of HGNN for Cora dataset, we trained with 5% of data, validated with 18.5% and tested with 37% of data. Table 8 shows the result. We have the following observations: **1)** The performance of Natural-HNN tends to be similar or slightly better than convolution-based models. This shows that Natural-HNN has good generalization power that is comparable to convolution-based methods. **2)** Our model performs better than recently introduced model, ED-HNN. Even if ED-HNN has much larger hyperparameter search space, Natural-HNN performs better due to generalization power.

# E Ablation studies and Hyperparameter sensitivity

## E.1 Selecting Alternative Branch

In Section 4, we used the representation earned from 'Disentangle-first Branch' ($h_{e_i}^k$) when creating final hyperedge factor representations ($\alpha_i^k h_{e_i}^k$). The experiment results below shows the result when using the other branch, 'Aggregation-first Branch' for creating final hyperedge factor representations ($\alpha_i^k \tilde{h}_{e_i}^k$). Table 9 shows the result for standard hypergraph benchmark dataset and Table 10 shows the result for cancer subtype classification task.

Table 9: Comparison of our model (first two rows) with alternative model that uses the other type of hyperedge factor representation (last two rows)

| Method | Cora | Citeseer | Pubmed | Cora-CA | DBLP-CA | NTU2012 | ModelNet40 | 20Newsgroups |
|---|---|---|---|---|---|---|---|---|
| Natural-HNN | 80.709 ± 1.635 | 73.285 ± 1.742 | 87.163 ± 0.450 | 84.993 ± 0.491 | 90.961 ± 0.137 | 89.900 ± 1.017 | 98.558 ± 0.295 | 81.734 ± 0.745 |
| Natural-HNN (+$\mathcal{L}_{dis}$) | 80.739 ± 1.570 | 73.551 ± 1.964 | 88.475 ± 0.466 | 85.081 ± 0.583 | 91.032 ± 0.179 | 90.060 ± 1.565 | 98.584 ± 0.254 | 81.827 ± 0.695 |
| Natural-HNN (other branch) | 80.650 ± 1.684 | 73.237 ± 1.678 | 87.137 ± 0.408 | 84.993 ± 0.434 | 90.968 ± 0.137 | 89.821 ± 0.847 | 98.557 ± 0.232 | 81.729 ± 0.701 |
| Natural-HNN (other branch + $\mathcal{L}_{dis}$) | 80.827 ± 1.157 | 73.575 ± 1.790 | 88.521 ± 0.424 | 85.081 ± 0.503 | 91.030 ± 0.178 | 90.060 ± 0.795 | 98.577 ± 0.227 | 81.837 ± 0.534 |

As we can see in Table 9, there is no big difference in the performance between using 'Disentangle-first Branch' and 'Aggregation-first Branch'.

Table 10: Comparison of our model (first row) with alternative model that uses the other type of hyperedge factor representation (last row).

| Method | BRCA | STAD | SARC | LGG | HNSC | CESC |
|---|---|---|---|---|---|---|
| Natural-HNN | 0.804 ± 0.036 | 0.659 ± 0.049 | 0.745 ± 0.045 | 0.707 ± 0.035 | 0.860 ± 0.042 | 0.881 ± 0.042 |
| Natural-HNN (other branch) | 0.797 ± 0.028 | 0.654 ± 0.041 | 0.747 ± 0.063 | 0.707 ± 0.033 | 0.863 ± 0.022 | 0.875 ± 0.051 |

As we can see in Table 10, there is no big difference in the performance between using 'Disentangle-first Branch' and 'Aggregation-first Branch'. The reason for this phenomenon is quite simple. We can consider the two cases: 1) when $h_{e_i}^k$ and $\tilde{h}_{e_i}^k$ are similar and 2) when they are largely different. **1)** When $h_{e_i}^k$ and $\tilde{h}_{e_i}^k$ are similar, the result will not differ a lot between using $h_{e_i}^k$ or $\tilde{h}_{e_i}^k$ as similar representations will be used. **2)** When $h_{e_i}^k$ and $\tilde{h}_{e_i}^k$ are largely different, the result will not be different a lot since relevance score $\alpha_i^k$ will be very small. In other words, $\alpha_i^k h_{e_i}^k - \alpha_i^k \tilde{h}_{e_i}^k = \alpha_i^k (h_{e_i}^k - \tilde{h}_{e_i}^k)$ will be very small for very small $\alpha_i^k$. This case means that the factor representation will not be reflected a lot during message passing since the representation is inconsistent (different result for two branches).

## E.2 Natural-HNN without naturality constraint

We performed another ablation study to check whether naturality condition proposed in the paper is important part that contributes to the model. We created an ablation model that do not satisfies naturality condition by not reflecting relevance score $\alpha_i^k$ during message passing. The results for standard hypergraph benchmark dataset is provided in Table 11. The results for the cancer subtype classification task are provided in Table 12.

Table 11: Model performance on standard hypergraph benchmark datasets (Accuracy). The ablation model does not satisfy the naturality condition.

| Method | Cora | Citeseer | Pubmed | Cora-CA | DBLP-CA | NTU2012 | ModelNet40 | 20Newsgroups |
|---|---|---|---|---|---|---|---|---|
| Natural-HNN (ours) | 80.709 ± 1.635 | 73.285 ± 1.742 | 87.136 ± 0.450 | 84.993 ± 0.491 | 90.961 ± 0.137 | 89.900 ± 1.017 | 98.558 ± 0.295 | 81.734 ± 0.745 |
| Natural-HNN (ours + $\mathcal{L}_{dis}$) | 80.739 ± 1.570 | 73.551 ± 1.964 | 88.475 ± 0.466 | 85.081 ± 0.583 | 91.032 ± 0.179 | 90.060 ± 1.565 | 98.584 ± 0.254 | 81.827 ± 0.695 |
| Natural-HNN (ablation) | 80.220 ± 1.573 | 73.237 ± 1.745 | 87.121 ± 0.170 | 84.874 ± 0.424 | 90.896 ± 0.165 | 89.281 ± 0.718 | 98.144 ± 0.226 | 81.685 ± 0.675 |
| Natural-HNN (ablation + $\mathcal{L}_{dis}$) | 80.250 ± 1.555 | 73.392 ± 1.832 | 88.448 ± 0.407 | 85.022 ± 0.508 | 90.968 ± 0.169 | 89.679 ± 1.129 | 98.177 ± 0.216 | 81.783 ± 0.771 |

In Table 11, we can see that there is a slight to moderate level of performance gap between Natural-HNN and its ablation model. It is not a surprising result that there is not big difference between them since standard benchmark datasets do not seem to have informative interaction contexts related to the task (Appendix D).

In Table 12, we can observe that there is a big difference between Natural-HNN and its ablation model. Since interaction context matters in cancer subtype classification task, naturality condition seems to boost the performance by capturing interaction context.

Table 12: Model performance on cancer subtype classification task (Macro F1). The ablation model does not satisfy the naturality condition.

| Method | BRCA | STAD | SARC | LGG | HNSC | CESC |
|---|---|---|---|---|---|---|
| Natural-HNN* (ours) | $0.804 \pm 0.036$ | $0.659 \pm 0.049$ | $0.745 \pm 0.045$ | $0.707 \pm 0.035$ | $0.862 \pm 0.045$ | $0.881 \pm 0.042$ |
| Natural-HNN* (ablation) | $0.756 \pm 0.031$ | $0.605 \pm 0.039$ | $0.713 \pm 0.071$ | $0.692 \pm 0.034$ | $0.814 \pm 0.037$ | $0.852 \pm 0.032$ |

## E.3 Hyperparameter Analysis

Since Natural-HNN does not have many hyperparameters, we analyzed how performance changes by the number of factors. Table 13 shows the result for the standard hypergraph benchmark dataset. Table 14 shows the result for cancer subtype classification task. Note that the tables below show the result of Natural-HNN without $\mathcal{L}_{dis}$.

Table 13: Performance of Natural-HNN with a different number of factors. The best performances (reported in Table 7) are marked in red.

| number of factors | Cora | Citeseer | Pubmed | Cora-CA | DBLP-CA | NTU2012 | ModelNet40 | 20Newsgroups |
|---|---|---|---|---|---|---|---|---|
| 1 | $80.384 \pm 1.820$ | $73.133 \pm 1.767$ | $87.063 \pm 0.373$ | $84.934 \pm 0.418$ | $90.951 \pm 0.139$ | $89.622 \pm 0.953$ | $98.480 \pm 0.310$ | $81.684 \pm 0.725$ |
| 2 | $80.532 \pm 1.638$ | $73.285 \pm 1.742$ | $87.055 \pm 0.401$ | $84.904 \pm 0.432$ | $90.961 \pm 0.137$ | $89.622 \pm 0.759$ | $98.513 \pm 0.272$ | $81.734 \pm 0.745$ |
| 4 | $80.709 \pm 1.652$ | $73.188 \pm 1.967$ | $87.083 \pm 0.450$ | $84.993 \pm 0.491$ | $90.939 \pm 0.151$ | $89.821 \pm 1.070$ | $98.558 \pm 0.295$ | $81.635 \pm 0.716$ |
| 8 | $80.591 \pm 1.673$ | $73.237 \pm 1.783$ | $87.136 \pm 0.450$ | $84.934 \pm 0.385$ | $90.955 \pm 0.131$ | $89.900 \pm 1.017$ | $98.513 \pm 0.286$ | $81.660 \pm 0.714$ |

We have interesting observations when we analyze the result in Table 7 with Table 13. **1)** In Table 7, we observe that Natural-HNN does not perform well on the Citeseer, Pubmed, and DBLP-CA datasets. Except for Pubmed, Table 13 shows that Natural-HNN used two or fewer factors on these datasets.

**2)** Natural-HNN demonstrated good performance on the remaining five datasets in Table 7. Except for the 20Newsgroups dataset, Natural-HNN used four or more factors to achieve its best performance, as shown in Table 13. These observations suggest that Natural-HNN generally performs well when capturing multiple factors. Furthermore, since the model did not benefit from using more than two factors on Citeseer and DBLP-CA, we suspect that these datasets lack diverse interaction contexts that would enhance performance. A similar trend is observed for other attention-based (AllSetTransformer) and disentanglement-based (HSDN) models in Table 7. Although these models are capable of capturing relational information, they showed poor performance—sometimes even worse than some convolution-based models.

Table 14: Performance of Natural-HNN with different number of factors. The best performance (reported in Table 1) are marked in red.

| number of factors | BRCA | STAD | SARC | LGG | HNSC | CESC |
|---|---|---|---|---|---|---|
| 1 | $0.789 \pm 0.036$ | $0.630 \pm 0.046$ | $0.729 \pm 0.055$ | $0.695 \pm 0.030$ | $0.853 \pm 0.047$ | $0.869 \pm 0.048$ |
| 2 | $0.787 \pm 0.038$ | $0.642 \pm 0.043$ | $0.745 \pm 0.045$ | $0.707 \pm 0.035$ | $0.858 \pm 0.031$ | $0.867 \pm 0.043$ |
| 4 | $0.804 \pm 0.036$ | $0.659 \pm 0.049$ | $0.725 \pm 0.048$ | $0.689 \pm 0.047$ | $0.858 \pm 0.036$ | $0.881 \pm 0.042$ |
| 8 | $0.785 \pm 0.027$ | $0.637 \pm 0.032$ | $0.729 \pm 0.058$ | $0.691 \pm 0.044$ | $0.860 \pm 0.042$ | $0.878 \pm 0.034$ |

We have similar observations when comparing the result in Table 1 and Table 14. **1)** For the SARC and LGG datasets in Table 14, Natural-HNN achieved its best performance when using two factors. **2)** For the remaining datasets, Natural-HNN achieved its best performance with four or more factors. Except for CESC, these cases showed a meaningful increase in performance. Therefore, we can draw a similar conclusion to the one derived from the comparison of Table 7 and Table 13.

# F  Additional Experiment Result

## F.1  Computational Complexity

Let $d_i$ be the input embedding dimension, $d_o$ be the output embedding dimension, $K$ be number of factors. $N$ denotes number of nodes and $M$ denotes number of hyperedges, $E$ denotes the number of node($v$)-hyperedge($e$) pair $(v, e)$ satisfying $v \in e$. We will assume that $d_i \geqslant d_o$, $d_o \geqslant K$, $E \geqslant M$ and $E \geqslant N$.

The computational complexity of one layer of Natural-HNN can be calculated by the following:

- Aggregation-first Branch (aggregation + MLP): $O(Ed_i) + O(Md_id_o)$
- Disentangle-first Branch (MLP + aggregation): $O(Nd_id_o) + O(Ed_o)$
- Similarity ($\alpha$) calculation : $O(K(\frac{d_o^2}{K^2} + \frac{d_o}{K})) = O(\frac{d_o^2}{K})$
- propagation back to nodes : $O(KE + Ed_o) = O(Ed_o)$
- other calculations (concat, interpolation by $\beta$) : $O(Nd_o)$ Thus, total computational complexity becomes $O((M + N)d_id_o + E(d_i + d_o + 1) + Nd_o + \frac{d_o^2}{K}) = O((M + N)d_id_o + E(d_i + d_o))$

For HGNN with dimension $d_i \geqslant d_e \geqslant d_o$ ($d_e$ denotes dimension of hyperedge embedding), computational complexity becomes $O(E(d_i + d_e) + (Md_i + Nd_o)d_e)$. The computational complexity of HGNN and Natural-HNN differs only by constant times. It is not surprising since Natural-HNN is quite similar to HGNN but instead use two branches (only) during Node-to-Hyperedge propagation and use factor similarity calculation. Thus, Natural-HNN is as scalable as HGNN.

## F.2  Scalability Analysis (training time)

We measured the time it takes for the model to train for 10 epochs. We averaged the values after measuring 5 times each. Also, we conducted the experiment in two settings: one with 2 heads and 16-dimensional vector as hidden representation and the other with 8 heads and 64-dimensional vector as hidden representation. Note that convolution-based models, AllDeepSets and ED-HNN (II) use 1 head as they do not have a multi-head attention mechanism. The table 15 shows the result of our model's scalability. We have the following observations: **1)** Our model is slower than convolution-based models and HSDN. Since convolution-based models use strong inductive bias with simple computations, they are naturally scalable than our model. HSDN took less time since they use only one message passing layer. **2)** Our model is much faster than all attention-based models. Thus, we can conclude that our model scales well with hypergraph and parameter size next to the convolution-based models.

Table 15: Time took for training 10 epochs for BRCA. We tested with two cases by differing hidden dimension size and number of heads†: multihead attention version

| ( dimension , head ) | (16, 2) | (64, 8)) |
|---|---|---|
| HGNN | $2.171 \pm 0.003$ | $8.492 \pm 0.010$ |
| HCHA | $2.130 \pm 0.003$ | $8.322 \pm 0.011$ |
| HNHN | $1.169 \pm 0.005$ | $4.362 \pm 0.005$ |
| UniGCNII | $2.384 \pm 0.004$ | $9.166 \pm 0.009$ |
| AllDeepSets | $7.870 \pm 0.026$ | $18.679 \pm 0.040$ |
| AllSetTransformer | $11.213 \pm 0.030$ | $27.004 \pm 0.024$ |
| HyperGAT† | $7.191 \pm 0.024$ | $24.579 \pm 0.047$ |
| SHINE† | $9.099 \pm 1.419$ | $22.253 \pm 0.162$ |
| HSDN | $2.944 \pm 0.003$ | $10.130 \pm 0.006$ |
| ED-HNN | $11.937 \pm 0.026$ | $22.738 \pm 0.026$ |
| ED-HNNII | $21.621 \pm 0.029$ | $36.418 \pm 0.026$ |
| Natural-HNN (ours) | $5.479 \pm 0.006$ | $18.924 \pm 0.070$ |

## F.3  Generalization power of Natural-HNN

To check the generalization power of our model, we experimented with different training set split ratio, while maintaining the validation and test set ratio to 25%. From 50%, we gradually reduced training set proportion to 10% as shown in Figure 17. Figure 17(a) and (b) are the result of measuring performance with accuracy or Macro-F1 scores and (c) and (d) are the result of measuring relative degradation of performance to the performance when trained with 50% For example, for BRCA dataset experiment, which is measured with Macro-F1 score, the relative degradation of performance is caculated by $(F_{50} - F_x)/F_{50} \times 100\%$ where $F_x$ denotes the Macro-F1 score when trained with x%. The same applies to Cora-CA, which is measured with accuracy. Figure 17 (a) and (c) are the result in Cora-CA dataset, which is standard hypergraph benchmark, (b) and (d) are the result

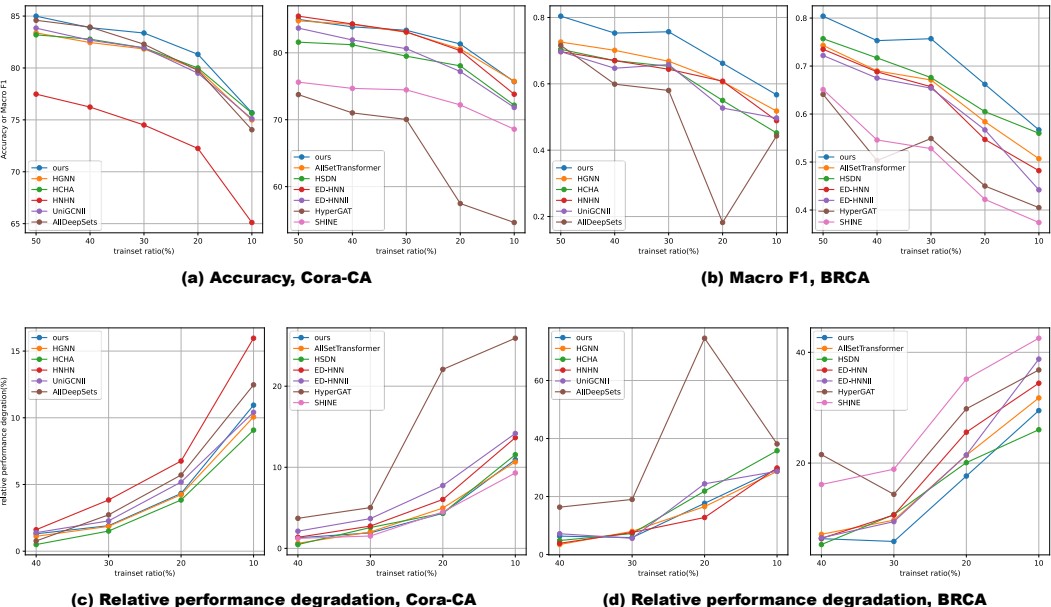

**(a) Accuracy, Cora-CA**

**(b) Macro F1, BRCA**

**(c) Relative performance degradation, Cora-CA**

**(d) Relative performance degradation, BRCA**

Figure 17: The performance of models when reducing training set proportion. First row shows Macro F1 score and the second row shows relative performance degradation compared to the performance when using 50% of dataset as training set. Natural-HNN (ours, colored in blue) maintains best Macro F1 score and small relative performance degradation on both Cora-CA and BRCA dataset.

for BRCA dataset, which is dataset used for cancer subtype classification task. The left figure in each Figure 17 (a,b,c,d) is the result of comparing ours (blue) and convolution of deepset based models. These baselines cannot perform context-dependent message passing. The right figure in each Figure 17 (a,b,c,d) is the result of comparing ours (blue) and other baselines that have potential for context-dependent message passing

We have the following observations : **1)** The degradation of performance for Natural-HNN was smaller when compared with most of the baselines in both Cora-CA and BRCA. Specifically, we can see that Natural-HNN has comparable result with convoluation based models in left figures of Figure 17 (c) and (d). Considering that convolutions based models have strong generalization performance due to their strong inductive bias, we can say that our model has good generalization power comparable to convolution based models. When compared with other baselinese in Figure 17 (b) and (d), we can observe that Natural-HNN had very small degradation in performance. In other words, Natural-HNN had nearly the smallest degradation when compared with models that have more expressive power than convolution based methods. We can consider our model had good generalization among baselines with more expressive powers. Specifically, in Figure 17 (d), Natural-HNN showed outstanding result in cancer dataset which has various context of interactions. This might be due to the fact that the inductive bias (context of interaction) that Natural-HNN used matched the actual data characteristics.

**2)** Natural-HNN had the best Macro-F1 score for all different training ratio. Our model always had the best performance compared to convolution or deepset based models in left figures of Figure 17 (a) and (b). Specifically, we can see that Natural-HNN had outstanding performance in BRCA cancer dataset in the left figure of Figure 17 (b). Thus, we can conclude that Natural-HNN is more expressive compared to convolution based models. Also, when inductive bias (interaction context) matches the data characteristics (BRCA), Natural-HNN provides outstanding performances. From the result, we could verify that Natural-HNN can utilize context information to get good performance. When compared with other baselines, in the right figures of Figure 17 (a) and (b), we can see that our model could achieve better, or at least comparable performance when compared with baselines. We can conclude that our model has expressive power comparable to other attention (including Set Transformer) or equivariance based models. Again, we can observe that Natural-HNN achieved outstanding performance in BRCA dataset by capturing context types. Considering that Natural-HNN had good generalization and expressivity, we argue that our model made a proper trade-off between expressive power and generalization.

## F.4 Captured Context in CESC

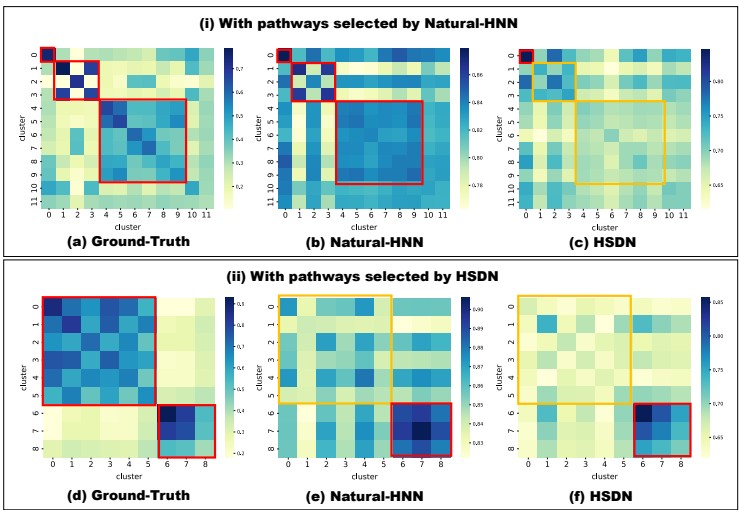

Figure 18: Captured interaction context. Pathways are selected by SHAP value. Captured patterns are shown in red box and not captured patterns are shown with orange box. Weakly captured case is marked as dotted red block.

Figure 18 shows the captured context result in CESC. The evaluation and interpretation method is identical to that of Section 5.3. As we can see in the figure, for pathways selected by Natural-HNN, Natural-HNN correctly captures context similarities between clusters (red box) while HSDN does not (orange box). For the pathways selected by HSDN, Natural-HNN and HSDN partially captures cluster similarity. However, when comparing orange box in (d) and (f), we can observe that Natural-HNN captures interaction context slightly better than HSDN even with the pathways selected by HSDN.

## F.5 Cancer Subtype Classification (Micro F1)

We briefly provide Micro F1 scores of each model in cancer subtype classification task. The Table 16 also shows that our model generally performs well on most of cancer datasets.

Table 16: Micro F1 score of each model with parameter and hyperparameter of the best Macro F1 score. Top two models are colored by **First**, **Second**. †: the variant of the model using multihead attention. ⋆ : we did not use $\mathcal{L}_{dis}$.

| Method | BRCA | STAD | SARC | LGG | HNSC | CESC |
|---|---|---|---|---|---|---|
| HGNN | 0.817 ± 0.027 | 0.727 ± 0.026 | 0.739 ± 0.057 | 0.696 ± 0.034 | 0.888 ± 0.031 | 0.903 ± 0.034 |
| HCHA | 0.808 ± 0.024 | 0.725 ± 0.036 | 0.731 ± 0.058 | 0.685 ± 0.039 | 0.876 ± 0.034 | 0.911 ± 0.034 |
| HNHN | 0.806 ± 0.027 | 0.729 ± 0.067 | 0.733 ± 0.046 | 0.676 ± 0.037 | 0.884 ± 0.018 | 0.910 ± 0.033 |
| UniGCNII | 0.791 ± 0.027 | 0.797 ± 0.038 | **0.761** ± 0.046 | 0.665 ± 0.038 | 0.910 ± 0.013 | 0.911 ± 0.018 |
| AllDeepSets | 0.823 ± 0.025 | 0.748 ± 0.039 | 0.657 ± 0.035 | 0.669 ± 0.045 | 0.895 ± 0.025 | 0.927 ± 0.024 |
| AllSetTransformer | 0.827 ± 0.031 | 0.710 ± 0.047 | 0.749 ± 0.047 | 0.656 ± 0.037 | 0.898 ± 0.016 | 0.908 ± 0.025 |
| HyperGAT | 0.754 ± 0.116 | 0.725 ± 0.050 | 0.645 ± 0.106 | 0.669 ± 0.051 | 0.889 ± 0.030 | 0.900 ± 0.025 |
| HyperGAT† | 0.753 ± 0.072 | 0.676 ± 0.108 | 0.643 ± 0.098 | 0.665 ± 0.042 | 0.883 ± 0.053 | 0.896 ± 0.021 |
| SHINE | 0.659 ± 0.090 | 0.590 ± 0.127 | 0.618 ± 0.106 | 0.649 ± 0.058 | 0.846 ± 0.032 | 0.890 ± 0.044 |
| SHINE† | 0.783 ± 0.027 | 0.711 ± 0.061 | 0.709 ± 0.045 | 0.654 ± 0.044 | 0.873 ± 0.027 | 0.907 ± 0.031 |
| HSDN | **0.838** ± 0.022 | **0.801** ± 0.033 | 0.758 ± 0.047 | 0.694 ± 0.036 | 0.892 ± 0.025 | 0.925 ± 0.024 |
| ED-HNN | 0.826 ± 0.024 | 0.793 ± 0.047 | **0.761** ± 0.039 | **0.703** ± 0.028 | 0.913 ± 0.021 | 0.925 ± 0.035 |
| ED-HNNII | 0.815 ± 0.027 | 0.748 ± 0.024 | 0.694 ± 0.050 | 0.696 ± 0.038 | **0.916** ± 0.013 | **0.942** ± 0.024 |
| Natural-HNN⋆ (ours) | **0.869** ± 0.024 | **0.824** ± 0.027 | **0.770** ± 0.040 | **0.709** ± 0.033 | **0.923** ± 0.020 | **0.932** ± 0.024 |

Table 17: Hyperedge classification result (accuracy). Top two models are colored by **First**, **Second**.

| Dataset | HGNN | HCHA | HNHN | UniGCNII | AllDeepSets | AllSetTransformer | HSDN | Natural-HNN (ours) |
|---|---|---|---|---|---|---|---|---|
| Chemical Reaction | 0.449 ± 0.005 | 0.482 ± 0.010 | 0.257 ± 0.008 | 0.672 ± 0.004 | 0.493 ± 0.023 | **0.727** ± 0.026 | 0.491 ± 0.023 | **0.773** ± 0.008 |

## F.6 Chemical Reaction Classification (Hyperedge Classification)

To validate whether Natural-HNN performs well not only on cancer subtype classification but also on other hypergraph datasets that contains meaningful hyperedge semantics, we performed hyperedge classification task on a chemical reaction dataset [21]. Among the three datasets proposed in that paper, we used the first dataset for validation, as the other two datasets have relatively small number of samples and the prediction tasks are too easy to serve as a meaningful evaluation of the model. The hyperparameter search space was kept the same as that used for the standard hypergraph benchmark datasets, and Natural-HNN was evaluated without $\mathcal{L}_{dis}$. As shown in Table 17, Natural-HNN demonstrates overwhelmingly superior performance compared to other models, including HSDN. Therefore, Natural-HNN proves to be highly effective not only for cancer subtype classification but also for datasets in which hyperedges contain hidden semantics related to labels.

## F.7 Reliability of Natural-HNN in Biology

In order for a model to be reliable, the model should provide consistent output regardless of the choice of hyperparameters. So we conducted an experiment to check whether models consistently rely on the same pathways. If a model consistently rely on the same pathways for prediction regardless of the hyperparameter, biologists might consider the model to be reliable since it potentially captured and used what can be explained with biological domain knowledge. On the other hand, if the model relies on different pathways for different hyperparameters, biologists might not trust the model.

To check whether model relies on the same pathways, we ranked the pathways with SHAP value and selected top-k pathways. These pathways are the ones that models relied most for their prediction. Then, we calculated Jaccard similarity of top-k pathways for different hyperparameters. If top-k pathways earned from each hyperparameter combination is similar, then we can conclude that model always rely on the same pathways regardless of the hyperparameters.

Figure 19 and Figure 20 are the result of calculating Jaccard similarity between different hyperparameter combinations on BRCA dataset. The hyperparameters we changed was the hidden dimension size and the number of factors. Values in each tick of row and column is the pair of the two hyperparameters (i.e., the value in the ticks represent (hidden dimesion, number of factors) pair). Each heatmap shows Jaccard similarity when selecting top 10, 15, 20, 50, 100 and 500 pathways. Figure 19 is the results for Natural-HNN and Figure 20 is the result for HSDN. We also calculated average Jaccard similarity for each heatmap.

The ideal result would show dark blue colors (high similarity) to all cells in the heatmap. It means that top-k pathways that a model relied on are always the same regardless of the hyperparameter. When comparing Figures 19 and 20, we can see that Natural-HNN tends to rely on the same pathway regardless of the hyperparameter while HSDN does not. When comparing average Jaccard similarity scores, we can quantitatively observe that Natural-HNN has better consistency when compared to HSDN. For example, Jaccard similarity with top 15 pathways of Natural-HNN (19 (b)) has average similarity of 0.759 while that of HSDN (20 (b)) has average similarity of 0.555.

From this experiment, we can conclude that Natural-HNN is reliable since it consistently focuses on the same pathways regardless of the choice of hyperparameters. Also, we could again verify that our model captures the functionality of pathways (interaction context of hyperedge) and expect that our model will work reliably in different dataset or different biological applications. Note that similar analysis for Figure 21 and Figure 22 provides similar conclusion.

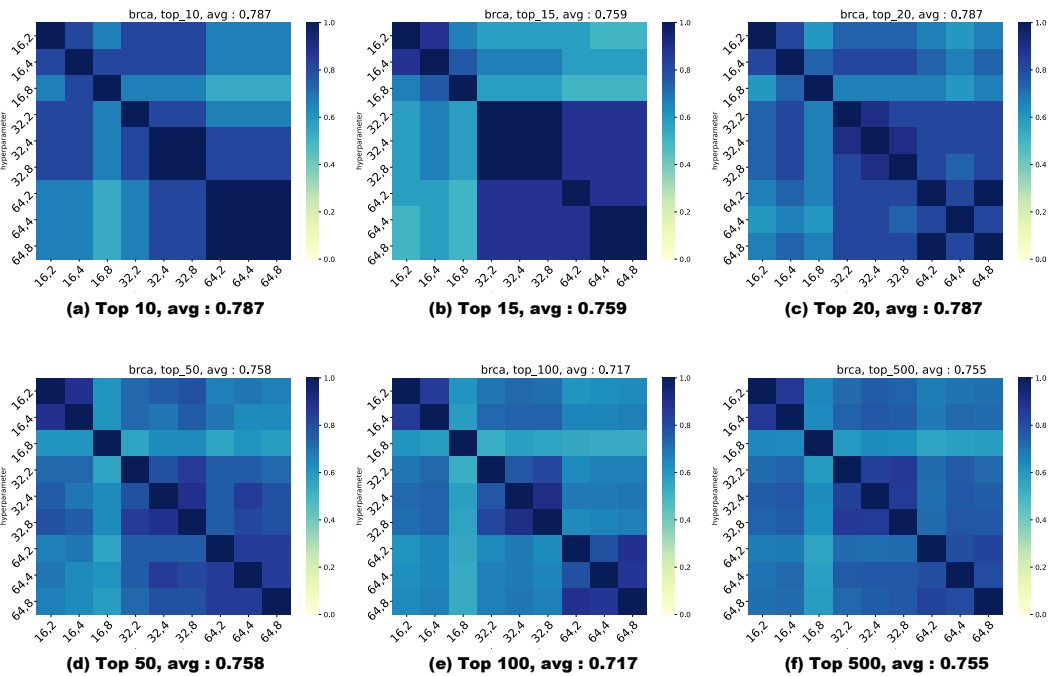

Figure 19: Jaccard similarity calculation result for Natural-HNN on BRCA. We can observe that Natural-HNN generally relies on similar pathways regardless of hyperparameters by showing high Jaccard similarity value.

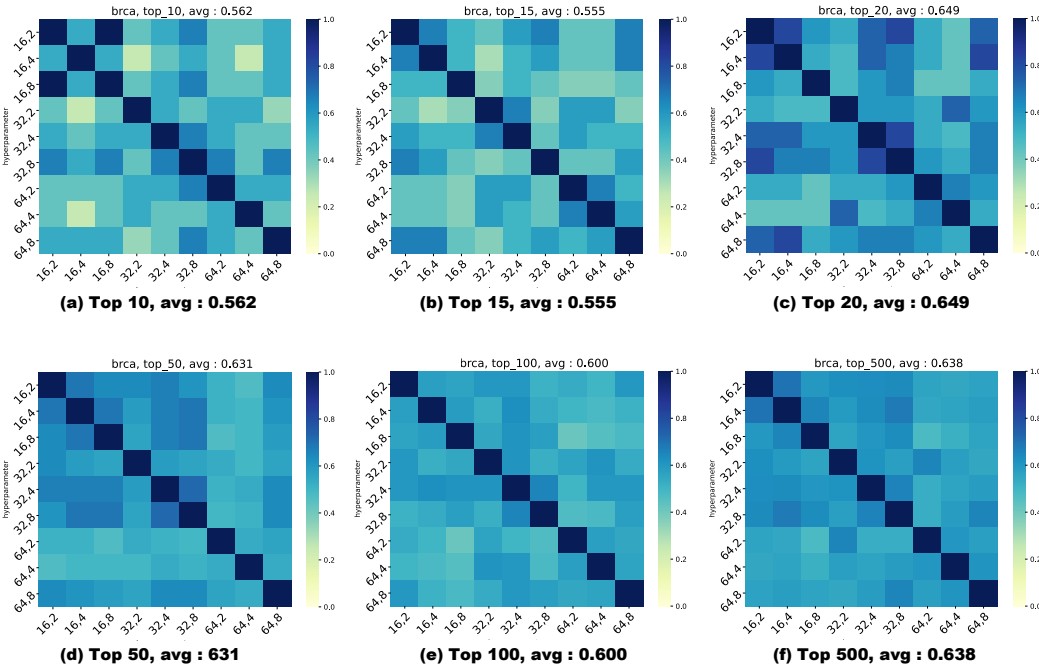

Figure 20: Jaccard similarity calculation result for HSDN on BRCA. We can observe that HSDN relies on different pathways for different hyperparameters by showing strong diagonal pattern. This inconsistency makes HSDN an unreliable model for biology.

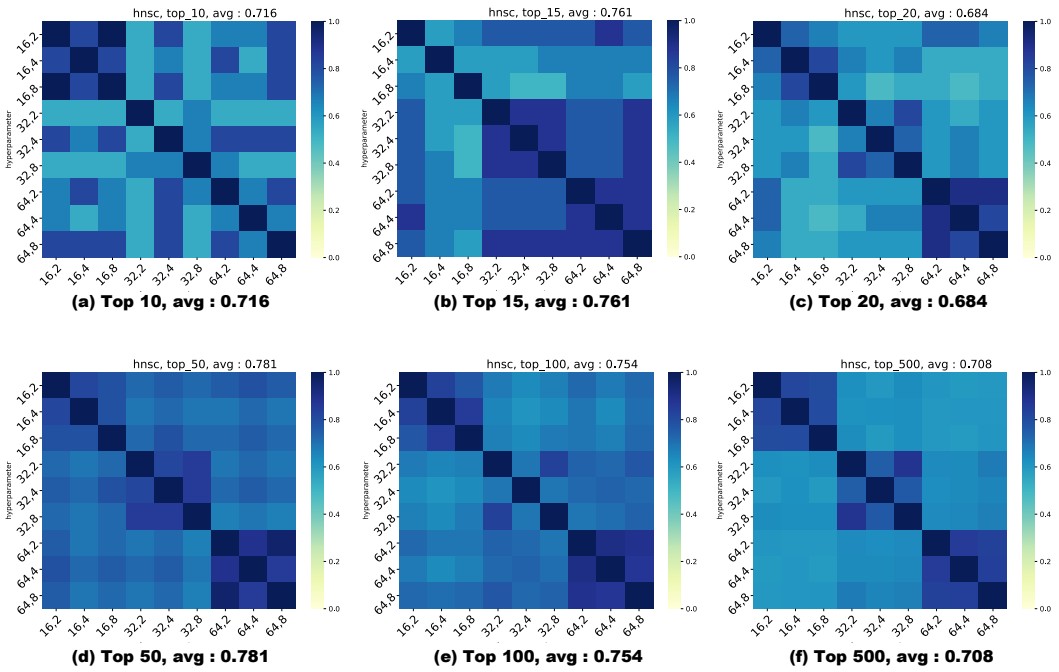

Figure 21: Jaccard similarity calculation result for Natural-HNN on HNSC. We can observe that Natural-HNN generally relies on similar pathways regardless of hyperparameters by showing high Jaccard similarity value.

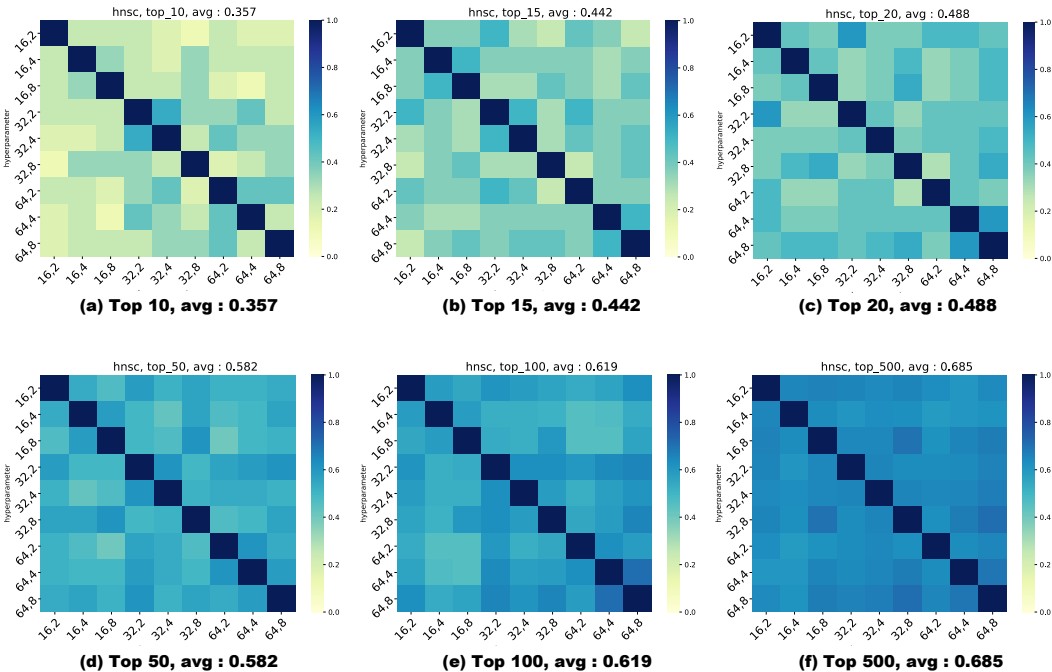

Figure 22: Jaccard similarity calculation result for HSDN on HNSC. We can observe that HSDN relies on different pathways for different hyperparameters by showing strong diagonal pattern. This inconsistency makes HSDN an unreliable model for biology.

# G  Limitations, impacts and Future Work

## G.1  Broader Impacts

**Potential Positive Societal Impacts.** As demonstrated through various experiments, Natural-HNN has the potential to capture the inherent heterogeneity of interactions and diverse interaction contexts. In complex systems such as biological organisms, many interactions have unknown functionalities. Natural-HNN's ability to capture these latent interaction contexts can contribute to the development of more reliable models for a wide range of real-world problems.

**Potential Negative Societal Impacts.** Our proposed method is designed to automatically identify and incorporate the factors underlying interactions. The relevance scores indicate which factors are most relevant to each interaction. However, if this method is applied to data where privacy is critical, it could potentially lead to indirect leakage of sensitive information through those relevance scores.

## G.2  Limitation

Natural-HNN uses hyperparameter $K$ to decide number of factors instead of automatically discovering the number of factors within data. In real world problems, it might require a lot of time to get optimal number of factors. This is a kind of a problem that all disentangle-based methods need to solve in the future.

## G.3  Future Work 1 : Model for Graph Neural Network

Since Natural-HNN is designed for hypergraph neural network, we can apply our model to graphs. However, it is computationally inefficient since Natural-HNN performs two step message passing (node-to-hyperedge, hyperedge-to-node) while most of the gnns perform one step message passing. Thus, we need to devise a novel criterion for disentangling edge types in graphs without using edge representations. Since there are many interaction types in graphs, developing reliable edge disentangling model in the perspective of category theory will be useful for many real world applications.

## G.4  Future Work 2 : Hyperedge-Node co-disentanglement

Our goal was to disentangle the factors behind group interactions, and thus we assumed that the nodes participating in an interaction share the same context (factor). However, it is also possible that individual nodes have their own distinct contexts (factors) when participating in an interaction. For example, consider a group discussion involving multiple individuals. In Natural-HNN, the disentanglement focused on hyperedge-level factors, such as the discussion topic. However, node-level disentanglement could also be applied in this scenario. Each participant might have a specific role in the discussion. Separately from the discussion topic, factors such as the context or role of each participant in the discussion could also be disentangled. Performing a hyperedge-node co-disentanglement, which is disentangling both hyperedge-level and node-level factors, would allow for a more nuanced approximation of diverse underlying mechanisms.

