# OpenReview forum: "Disentangling Hyperedges through the Lens of Category Theory"
_NeurIPS.cc/2025/Conference — NeurIPS 2025 poster_

### Official Review · Reviewer_DgXL · 2025-06-27

**Clarity:** 2
**Significance:** 3
**Originality:** 3
**Rating:** 4
**Confidence:** 4

**Summary:**

The paper introduce factor-representation consistency, a new criterion to find hidden factors inside hyperedges. This criterion comes from category theory: if we first disentangle nodes then aggregate, result should be same as first aggregate then disentangle. Authors build a model named Natural-HNN that implements this rule as an extra loss. They evaluate the peformance of the model on six cancer hypergraph datasets and show that the method performs better that baselines. They also show that the learned factors match known gene-pathway functions.

**Questions:**

Q1 – Could you add, in Section 1 or a new “Problem Definition” part, formal explanations of (a) what exactly you call disentanglement in hypergraphs and (b) what a factor is (latent variable, dimension in the embedding, etc.)?

Q2 – Beyond empirical evidence, can you give a theoretical or synthetic argument showing why the equality between “aggregate → disentangle” and “disentangle → aggregate” is expected when factors are truly causal for a hyperedge? Even a toy proof or a controlled synthetic experiment would strengthen the idea.

Q3 – Domain generalisation evidence
All experiments are biomedical, while e.g. HSDN was tested on several domains (social, recommendation, citation). This makes it hard to judge whether your approach really generalises beyond genomics. Could you add at least one public non-biomedical hypergraph benchmark, e.g., DBLP-coauthorship, MovieLens, Yelp, or explain why such transfer is not possible? Showing competitive performance outside the cancer datasets would greatly increase the perceived significance of the method.

Q4 – Table 7 hints at a variant of Natural-HNN trained without the consistency loss. Could you move this comparison to the main results, report exact numbers, and discuss the gap? This will clarify how much improvement comes from the new criterion itself.

**Ethical Concerns:**

["NO or VERY MINOR ethics concerns only"]

**Final Justification:**

While I am satisfied with some of the responses, I believe the clarity issues raised in W1/Q1 are not easily addressed without a more substantial restructuring or rewriting of parts of the paper. I consider this a good paper overall, but given the outstanding concerns, I will maintain my current rating for now and await further discussion and the conclusion of the discussion period.

**Limitations:**

The paper does include a dedicated limitations section in Appendix, but it largely reiterates compute cost and future work; it should explicitly discuss domain specificity and absence of theoretical guarantees.

**Paper Formatting Concerns:**

no formatting concerns

**Quality:**

3

**Strengths And Weaknesses:**

Strengths
S1: Good technical quality. The naturality-based loss is clearly derived, and experiments use 10 random splits plus useful ablation studies.
S2: Originality. To the best of my knowledge, this is the first work that applies the naturality concept from category theory to hyperedge disentanglement.
S3: On six TCGA cancer datasets the method improves macro-F1 by about 4–8 pp over twelve strong baselines.
S4: Implementation details, hyper-parameters, and anonymous code are provided.

Weaknesses.
W1: Key concepts are not defined. Section 1 never explains what “disentanglement” means in the hypergraph setting. The term “factor” appears many times but a formal definition is missing, so the reader can only guess that it refers to some latent variables explaining hyperedges.
W2: Naturality lacks theoretical support. The paper shows that the naturality criterion works in experiments, but it first needs a higher-level justification—why should consistency between “aggregate-then-disentangle” and “disentangle-then-aggregate” reveal true factors?
W3: Domain scope is narrow. All experiments come from one biomedical domain; there is no evidence that the approach generalises to social, recommendation, or synthetic hypergraphs.

---

> ### Author Rebuttal · Authors · 2025-07-31
>
> # Q1 & W1
> Disentanglement refers to the process of decomposing an entity into multiple independent latent components, or factors. For example, a feature vector can be projected into several distinct latent spaces, each capturing a different aspect of the underlying information, resulting in a set of factor representations.
>
> In the context of hyperedge-level disentanglement, this involves decomposing a hyperedge into multiple types (factors), each corresponding to a distinct interaction mechanism or condition. We will add the explanation in the final version of the paper.
>
> # Q2 & W2
> Let us begin by analyzing the definition of hyperedge disentanglement in category theoretical viewpoint. Let $V$ be a set of nodes in a hyperedge and let $E$ be the result of group interaction (intuitively, conclusion of group discussion). The transition from $V$ to $E$ is the group interaction mechanism. In category theory, we can represent this as morphism $f$ : $V \rightarrow E$.
>
> By the definition of hyperedge disentanglement, mechanism can be decomposed into several independent mechanisms. (i.e., morphism $f$ can be decomposed as the product of independent morphisms $f_{c}$ and $f_{d}$. $f = f_{c} \times f_{d}$)
> As can be seen in Figure 13 at the Appendix A.5, objects (nodes $V$ and conclusion $E$) can be decomposed correspondingly : $V = V_{c} \times V_{d}$ and $E = E_{c} \times E_{d}$ such that $f_{c} : V_{c} \rightarrow E_{c}$ and $f_{d} : V_{d} \rightarrow E_{d}$.
>
> Now, let’s examine this from the perspective of factor $c$. We have the commutative diagram for factor $c$. Disentangle first ($V \rightarrow V_{c}$) and then performing interaction $ f_{c} : V_{c} \rightarrow E_{c} $ results $E_{c}$.
> Performing interaction $f : V \rightarrow E$ first, and then disentangling $E \rightarrow E_{c}$ results $E_{c}$.
> This is category-theoretical representation of hyperedge disentanglement.
>
> Now, let’s bring this abstract concept into the representation space. In category theory, these mappings are carried out through a functor. Functor $F$ will map entities in the definition into representation spaces.
> Since functors are structure-preserving maps between different domains, the commutative diagram will also hold in the representation space.
>
> Disentangle and then message passing : $ F(V) \rightarrow F(V_{c}) \rightarrow F(E_{c})$
>
> Message passing and then Disentangling : $ F(V) \rightarrow F(E) \rightarrow F(E_{c})$
>
> Thus, regardless of the sequence of action, we get the same hyperedge factor representation $F(E_{c})$.
>
> Let’s now explain this concept through a more intuitive example. Consider a group discussion involving several people. In this scenario, each factor represents a discussion topic. Suppose there are two independent discussion topics, $c$ and $d$, that do not influence each other. Whether we discuss both topics and then extract the conclusion for $c$, or we focus on topic $c$ alone to derive the conclusion, the conclusion of discussion for $c$ will be the same. This is because $c$ and $d$ are independent, and thus do not affect one another.
>
>
>
> # Q3 & W3
> The results on the hypergraph benchmark datasets in Appendix D include citation networks such as DBLP-coauthorship. For disentanglement to lead to substantial performance improvement, the underlying context or factors must be strongly associated with the labels. If the context exists but is only weakly related to the labels, even a well-disentangled representation may not yield a significant performance gain.
>
> For example, in co-citation and co-authorship networks, hyperedges are formed by grouping all documents cited by a single paper or all papers authored by a single researcher. While citations between a pair of papers may carry meaningful context (e.g., a specific reason for citation), it is difficult to argue that a group of cited documents shares a well-defined or coherent context. Even if we assume that some interaction context exists in such hyperedges, the connection between this context and node labels is often unclear. In such cases, even if contextual information is captured, it may not contribute meaningfully to label prediction performance.
>
> In contrast, in the cancer subtype classification task, pathways are known to carry functional context [2,3] and are closely associated with the labels (i.e., cancer types) [2,4]. To summarize briefly, pathways are associated with specific biological functions (i.e., factors), and whether these functions operate normally—potentially reflected in the factor representations—can determine the type of disease. This indicates that, in this case, the factors are highly relevant to the labels.
>
> $\textbf{To assess the applicability of our model beyond genomics}$, we conducted a hyperedge classification experiment on the Chemical Reaction Dataset 1 [1]. In this task, the goal is to classify the reaction type (i.e., the hyperedge label) given a set of molecules (nodes) participating in the reaction.
>
> Due to time constraints, we selected a subset of representative baseline models to evaluate their performance in terms of Macro-F1 score. The results are summarized in the table below. We will evaluate the remaining baseline models and include their performance results in the final version of the paper.
> |                                        | HGNN        | HNHN        | AllDeepSets   | AllSetTransformer | HSDN        | **Natural-HNN (ours)** |
> |----------------------------------------|-------------|-------------|---------------|-------------------|-------------|--------------------|
> | Molecule Reaction Dataset 1 (Macro-F1) | 0.449±0.005 | 0.257±0.008 |  0.493±0.023 |  0.727±0.026     | 0.491±0.023 | **0.773±0.008**        |
>
> In the chemical reaction dataset, Natural-HNN shows a large performance gap compared to HSDN, which performs hypergraph-structure level disentanglement, and also outperforms AllSetTransformer by approximately 5 percentage points. These results suggest that naturality-based disentanglement can be effectively applied not only in biomedical domain but also in other domains.
>
> # Q4
> Table 7 presents the performance with and without the factor discrimination loss introduced in Section 4.4. The comparison based on the presence or absence of the naturality constraint (consistency) was shown in Table 11. We will incorporate these results into the main table and add corresponding explanations in the final version of the paper.
>
>
>
> [1] DLGNet: Hyperedge Classification through Directed Line Graphs for Chemical Reactions (OpenReview)
>
> [2] Mapping biological process relationships and disease perturbations within a pathway network.
>
> [3] Disentangling the multigenic and pleiotropic nature of molecular function
>
> [4] Identifying cellular cancer mechanisms through pathway-driven data integration

---

> > ### Comment · Reviewer_DgXL · 2025-08-04
> >
> > I thank the authors for providing a detailed rebuttal. While I am satisfied with some of the responses, I believe the clarity issues raised in W1/Q1 are not easily addressed without a more substantial rewriting of parts of the paper. I consider this a good paper overall, but given the outstanding concerns, I will maintain my current rating for now and await further discussion and the conclusion of the discussion period with other reviewers.

---

> > > ### Author Response · Authors · 2025-08-04
> > >
> > > Thank you for your positive feedback on our paper.
> > >
> > >  We apologize for not fully addressing your concern in W1/Q1 in our previous response.
> > >  In $\textbf{Discussion Part 1}$, we will discuss the formal definitions of the terms, and in $\textbf{Discussion Part 2}$, we will cover the problem definition.
> > >
> > > $\textbf{1) Discussion Part 1}$
> > >
> > > In general, the term "factor" refers to an $\textbf{abstract concept that corresponds to a latent pattern}$ hidden within the data. Since the concept of a factor is inherently abstract, it is more common to explain it through examples rather than providing a strict formal definition.
> > >
> > >  For example, in graph edge disentanglement, different semantics of edges are treated as factors. An example could be the relationships between individuals in a social network.
> > >  In graph structure disentanglement, the semantics of substructures are considered as factors. An example would be the functional substructures in a chemical graph.
> > >
> > > In my work, hyperedge disentanglement treats the independent semantics of hyperedges that are relevant to labels as factors. A closely related concept is the (hidden, not explicitly provided in the dataset) hyperedge type. Examples include the functionality of pathways or the discussion topics in a group discussion.
> > >
> > >  The most formal description ‘ without equations ’ would be: “Hyperedge disentanglement assumes that group interactions (i.e., hyperedges) have mechanisms that determine the labels, and aims to capture the factors (i.e., context or condition) that influence these group interactions.” (Introduction line 32-34)
> > > Although the definition of the term 'factor' in disentanglement is somewhat abstract, problem definition can be rigorously formulated through equations.
> > >
> > > $\textbf{2) Discussion Part 2}$ : Problem Definition
> > >
> > >  To better address your concern, we added notations and wrote the problem definition as follows:
> > >
> > > (As described in the $\textbf{Notation}$ in Section 3, $\mathcal{V} = \\{ v_{1}, v_{2}, ... , v_{N} \\} $ indicates nodes, $\mathcal{E} = \\{e_{1}, e_{2}, ... , e_{M}\\}$ indicates hyperedges and $X = \\{x_{v_{1}}, ..., x_{v_{N}} \\}$ indicates given node features.)
> > >
> > > $\textbf{Additional Notation}$:
> > >
> > > We additionally denote ground truth values or labels of nodes as $ L = \\{l_1, …, l_N \\} $.
> > > Incidence matrix $\mathcal{I} \in \\{ 0,1 \\}^{N \times M}$ indicates whether a node belongs to a hyperedge or not.
> > > In hyperedge disentanglement, there is a set of disentangled incidence matrix $\mathcal{I}^{dis} = \\{ \mathcal{I}_1, …, \mathcal{I}_K \\}$ where $\mathcal{I}_i \in \\{ 0,1 \\}^{N \times M}$ denotes incidence matrix of subhypergraph for factor $i$. Note that the ground truth $\mathcal{I}^{dis}$ exists, but is not given in dataset.
> > >
> > > $\textbf{Problem Definition}$:
> > >
> > > Let’s assume there is a ground truth function or mechanism $f$ that maps features to labels through interactions (i.e., $ L = f ( \mathcal{I}, X ) $).
> > > This ground truth mechanism is composed of $K$ independent mechanisms $f_1, …, f_K$. In other words, $f(\mathcal{I}, X) = \bigoplus_{ i=1,...,K }  f_i ( \mathcal{I}_i ,  X) $ where $\bigoplus$ denotes concatenation.
> > >
> > > The objective of hyperedge disentanglement is to approximate ground truth mechanism $f$ with disentangled-hypergraph neural network $f'(\mathcal{I}, X) = \bigoplus_{ i=1,...,K }  f'_i ( \mathcal{I} , X ) $.
> > > Each factor-specific component of a model $f’_i ( \mathcal{I} , X ) $ should be able to internally infer $ \mathcal{I}'_i$  that is similar to ground truth $ \mathcal{I}_i$ and then approximate $f_i( \mathcal{I}_i ,  X )$ from $X$ and$ \mathcal{I}'_i $. That is, $f'_i( \mathcal{I} ,X) =  g_i(h_i( \mathcal{I} , X), X) \sim f( \mathcal{I}_i, X )$ such that $ h_i( \mathcal{I}, X) \sim \mathcal{I}_i $
> > >
> > > Contents in discussion part 1 will be briefly added to Section 2.1 and the contents in discussion part 2 will be added right before Section 3.1
> > >
> > > We are open to further addressing your concern during the remaining discussion period.
> > >  If the explanation above still does not sufficiently address your concern, we would appreciate it if you could let us know which aspects you feel are still lacking.

---

### Official Review · Reviewer_28h3 · 2025-07-01

**Clarity:** 2
**Significance:** 2
**Originality:** 3
**Rating:** 4
**Confidence:** 4

**Summary:**

This paper addresses the issue of hyperedge disentanglement in hypergraph neural networks (HNNs). The authors propose a new criterion for hyperedge disentanglement based on category theory, with a particular focus on the naturality condition. Their approach dissects hyperedges within hypergraphs, paying special attention to meaningful factors such as functional context. To validate the proposed criterion, the authors implement a proof-of-concept model—Natural-HNN—and apply it to a cancer subtype classification task using genetic pathway data. The experimental results demonstrate that the model outperforms traditional baseline models, thereby validating the effectiveness of the proposed disentanglement criterion.

**Questions:**

1. Could the authors provide further clarification on the premise mentioned in Weakness 1?

2. Could the authors provide a more detailed explanation of how the method performs hyperedge disentanglement?

3. The improvements in the method do not seem very significant, but the visualization experiments appear to be very effective. Could the authors explain why this is the case?

**Ethical Concerns:**

["NO or VERY MINOR ethics concerns only"]

**Final Justification:**

The authors' rebuttal has addressed my concerns. I believe this paper is worthy of acceptance.

**Limitations:**

yes

**Paper Formatting Concerns:**

According to the paper format, the appendix should be attached at the end of the main PDF, rather than being placed in the supplementary file.

**Quality:**

3

**Strengths And Weaknesses:**

Strengths
1. This work introduces a new category-theoretical perspective on hyperedge disentanglement, which is rather novel in the context of hypergraph learning.
2. The paper clearly identifies its contributions, including the novel criterion for disentangling hyperedges, the Natural-HNN model, and experimental validation on cancer subtype classification.

weakness
1. The proposed method in the paper is based on the premise: "The factor representation must be consistent regardless of the sequence of operations if that factor is relevant to the interaction context of a hyperedge." However, this premise has not been convincingly proven theoretically, and the explanation provided is not sufficiently detailed. Is it possible that the representations could differ? Personally, I believe this is indeed a possibility. The representation of the same target does not necessarily have to be fixed; perhaps new information emerges after hyperedge disentanglement, or the neural network has modeled different relationships. This phenomenon is quite likely to occur in connectionist models like hypergraph neural networks. I think this point should at least be discussed in more detail, and more space should be devoted to this argument.
2. The specific process of hyperedge disentanglement is not clearly elaborated, which should be one of the key focuses of the paper.

---

> ### Author Rebuttal · Authors · 2025-07-31
>
> # Q1 & W1
> Disentanglement is a methodology based on the assumption that data can be decomposed into independent factors, and it aims to identify and isolate these underlying factors. Importantly, the disentanglement process does not introduce new information; rather, it extracts specific factor-related information from the existing data.
>
> Consider the example of people (nodes) participating in group discussions (hyperedges). A node representation corresponds to an individual's opinion. Suppose each factor represents an independent discussion topic (e.g., Factor 1: "Is Hawaiian pizza tasty?" Factor 2: "Do you support a particular political party?"). Then, the factor representation of a node would reflect that individual’s stance on the corresponding topic. In this context, disentanglement refers to the process of extracting the relevant opinions from each person’s overall set of beliefs, focusing only on a specific topic. This extraction does not create new meanings but selectively reveals components of existing information.
>
> Now, consider that a group discussion eventually leads to a conclusion—this can be seen as the hyperedge representation. An entangled hyperedge representation may be thought of as a comprehensive outcome resulting from a discussion that simultaneously covers all possible topics. However, if the topics (factors) are independent of one another, extracting the conclusion related to a specific topic from an entangled hyperedge representation (i.e., message passing and then disentangle) yields the same result as conducting a discussion focused solely on that topic (i.e., disentangle and then message passing).
>
> To make this more concrete, suppose a group discusses two independent topics: Factor 1, "Is Hawaiian pizza tasty?" and Factor 2, "Do you support a particular political party?" These topics are assumed to be independent—opinions on one do not influence opinions on the other. Therefore, the conclusion reached about political party preference after a discussion covering both topics (message passing followed by disentanglement) will be equivalent to the conclusion reached after a discussion focused solely on political preference (disentanglement followed by message passing).
>
> In fact, this is not merely a premise but a fundamental property that must always hold.
> Let us begin the proof by analyzing the definition of hyperedge disentanglement in category theoretical viewpoint. Let $V$ be a set of nodes in a hyperedge and let $E$ be the result of group interaction (intuitively, conclusion of group discussion). The transition from $V$ to $E$ is the group interaction mechanism. In category theory, we can represent this as morphism $f$ : $V \rightarrow E$.
>
> By the definition of hyperedge disentanglement, mechanism can be decomposed into several independent mechanisms. (i.e., morphism $f$ can be decomposed as the product of independent morphisms $f_{c}$ and $f_{d}$. $f = f_{c} \times f_{d}$)
> As can be seen in Figure 13 at the Appendix A.5, objects (nodes $V$ and conclusion $E$) can be decomposed correspondingly : $V = V_{c} \times V_{d}$ and $E = E_{c} \times E_{d}$ such that $f_{c} : V_{c} \rightarrow E_{c}$ and $f_{d} : V_{d} \rightarrow E_{d}$.
>
> Now, let’s examine this from the perspective of factor $c$. We have the commutative diagram for factor $c$. Disentangle first ($V \rightarrow V_{c}$) and then performing interaction $ f_{c} : V_{c} \rightarrow E_{c} $ results $E_{c}$.
> Performing interaction $f : V \rightarrow E$ first, and then disentangling $E \rightarrow E_{c}$ results $E_{c}$.
> This is category-theoretical representation of hyperedge disentanglement.
>
> Now, let’s bring this abstract concept into the representation space. In category theory, these mappings are carried out through a functor. Functor $F$ will map entities in the definition into representation spaces.
> Since functors are structure-preserving maps between different domains, the commutative diagram will also hold in the representation space.
>
> Disentangle and then message passing : $ F(V) \rightarrow F(V_{c}) \rightarrow F(E_{c})$
>
> Message passing and then Disentangling : $ F(V) \rightarrow F(E) \rightarrow F(E_{c})$
>
> Thus, regardless of the sequence of action, we get the same hyperedge factor representation $F(E_{c})$.
>
> # Q2 & W2
> The criterion we derived from category theory establishes that performing message passing followed by disentanglement (disentanglement is implemented as MLP), or disentanglement followed by message passing, should yield consistent results. As demonstrated in our proof-of-concept model, when the hyperedge factor representations obtained through these two paths are similar, we interpret this as the corresponding factor being highly relevant to the hyperedge. Consequently, we assign a higher weight ($\alpha_{i}^{k}$) to that factor, allowing it to contribute more significantly to the final representation.
>
> # Q3
> The naturality-based criterion is derived directly from the definition of hyperedge disentanglement. By guiding the disentanglement process based on this property, our model was able to capture factors that are closely related to the functional context (i.e., hyperedge context), as clearly demonstrated in the visualizations. Because the proposed criterion effectively captures contextual information, our model achieved the best performance on all datasets except CESC. Notably, on four of the datasets, it outperformed the second-best model by a margin of more than 1.5 point (up to 4.7 point), further highlighting its effectiveness.

---

> ### Author Response · Authors · 2025-08-07
>
> Dear reviewer 28h3,
>
> We kindly request you to review our rebuttal, as the author-reviewer discussion period is nearing its end.
> We hope to know whether our rebuttal adequately addressed your concerns. Let us know if you have any remaining questions.

---

> > ### Comment · Reviewer_28h3 · 2025-08-07
> > **Reply**
> >
> > Sorry for the late reply. Initially, I thought that if the algorithm you used were formally and provably correct, then the conclusions drawn from it would also be unquestionably valid. However, since your method is based on neural networks, I was concerned that this might introduce potential issues. That said, I now understand that your approach is actually trained under this assumption as a premise. I will revise my score accordingly. I also hope that this part can be discussed in greater detail in the paper.

---

> > > ### Author Response · Authors · 2025-08-07
> > >
> > > Thank you for your response and valuable feedback! We will address the point you raised by adding a dedicated paragraph to Section 2.1 in the final version.

---

### Official Review · Reviewer_gAsC · 2025-07-02

**Clarity:** 2
**Significance:** 3
**Originality:** 3
**Rating:** 4
**Confidence:** 3

**Summary:**

This paper addresses hyperedge disentanglement problem in hypergraph neural networks. While previous work has focused on disentangling node or edge-level representations, the authors argue that group-level interactions (hyperedges) encode rich semantic factors,especially in domains like biology, adn drug discovery, where hyperedges represent pathways or functional modules.
The authors propose a novel disentanglement criterion based on category theory, which ensures that the representation of a factor is consistent regardless of whether disentanglement is applied before or after message passing. They introduce Natural-HNN, which computes two versions of hyperedge factor representations and measures their agreement to assess factor relevance.

**Questions:**

- Can you clarify why category theory is necessary for defining the disentanglement criterion, and why alternative heuristic or empirical approaches would fall short?

- Have you considered evaluating disentanglement using quantitative metrics such as mutual information, total correlation, or latent factor separation?

- Could the naturality-based criterion be integrated into simpler or more conventional GNN frameworks, such as bipartite models?

**Ethical Concerns:**

["NO or VERY MINOR ethics concerns only"]

**Limitations:**

yes

**Quality:**

3

**Strengths And Weaknesses:**

## Strenghts

- Hyperedge disentanglement is an unexplored and novel problem.
- Natural-HNN consistently outperforms other baselines across multiple datasets.
- The model’s disentangled factors offer interpretable insights into biological pathway functions

## Weaknesses
- It is not clear why this problem must be addressed using category theory; the motivation for this choice needs to be made stronger in the Introduction. As it stands, it is unclear whether simpler methods could achieve similar outcomes without the added abstraction.

- All experiments focus on biomedical hypergraphs. The common hypergraph datasets used in literature has not been explored.
- The evaluation of disentanglement relies on SHAP and functional similarity metrics rather than direct ground truth or quantitative disentanglement measures. Given that disentanglement is the paper’s core contribution, a more rigorous and targeted evaluation is necessary.

---

> ### Author Rebuttal · Authors · 2025-07-31
>
> # Q1 & W1
> Heuristic-based methods typically rely on assumptions about the data. While such methods may perform well when those assumptions hold, they often struggle when the assumptions are violated. A representative example is the similarity-based criterion, which assumes that instances belonging to the same factor will have similar factor representations. However, this assumption does not always hold in practice. For instance, in genetic pathways, genes within the same pathway or context do not necessarily share similar characteristics or gene expression values.
>
> One of the baseline models, HSDN, adopts a similarity-based criterion. As shown in Figure 5, it failed to disentangle hyperedges effectively, likely due to the limitations of its heuristic assumptions.
>
> In contrast, our goal was to develop a criterion for hyperedge disentanglement that does not rely on heuristics. To achieve this, we sought to identify properties that can be directly derived from the definition of hyperedge-level disentanglement itself. As an analytical tool, we employed category theory, as it provides a mathematically rigorous framework for representing abstract concepts and complex systems.
>
> By formulating hyperedge disentanglement within the language of category theory, we were able to formally express its underlying structure and derive a principled criterion that facilitates effective disentanglement.
> We will incorporate this motivation into the final version of the paper.
>
>
> # Q2 & W3
> Since there is no ground truth for pathway functions, it is not possible to directly compare the learned factors with ground-truth labels. Even if such labels existed, meaningful evaluation would still be difficult without a guarantee that the pathway functions are independent—because disentanglement fundamentally aims to uncover independent factors.
>
> However, evaluating hyperedge-level disentanglement is feasible. Although explicit function labels are unavailable, functional similarity between pathways can be calculated using widely adopted methods in computational biology. By comparing functional similarity from computational biology method and that of ours, we were able to verify that our model captures hyperedge-level factors, rather than hypergraph-structure-level or other types of disentanglement.
>
> Furthermore, as shown in Figure 6, we analyzed the correlation between hyperedge-level factors. This analysis provides supporting evidence that each factor captures distinct information or mechanisms.
>
>
> # Q3
> We believe that our approach can also be applied to bipartite GNNs. Our model essentially consists of two phases: node-to-hyperedge message passing and hyperedge-to-node message passing. If nodes and hyperedges are regarded as two distinct node types, the structure can be interpreted as a bipartite graph. Therefore, the proposed naturality-based criterion is also applicable in the context of bipartite GNNs.
>
> # W2
> The results on the standard hypergraph benchmark datasets (including citation networks) are provided in Appendix D. Although we report performance on the benchmark datasets, it is difficult to conduct in-depth analysis comparable to that of the cancer subtype classification task, as the underlying factors present in these benchmark datasets are not clearly defined or known.
>
> Additionally, during the rebuttal period, we conducted a hyperedge classification experiment on the Chemical Reaction Dataset 1 [1]. In this task, the goal is to classify the reaction type (i.e., the hyperedge label) given a set of molecules (nodes) participating in the reaction.
>
> Due to time constraints, we selected a subset of representative baseline models to evaluate their performance in terms of Macro-F1 score. The results are summarized in the table below. We will evaluate the remaining baseline models and include their performance results in the final version of the paper.
> |                                        | HGNN        | HNHN        | AllDeepSets   | AllSetTransformer | HSDN        | **Natural-HNN (ours)** |
> |----------------------------------------|-------------|-------------|---------------|-------------------|-------------|--------------------|
> | Molecule Reaction Dataset 1 (Macro-F1) | 0.449±0.005 | 0.257±0.008 |  0.493±0.023 |  0.727±0.026     | 0.491±0.023 | **0.773±0.008**        |
>
> In the chemical reaction dataset, Natural-HNN shows a large performance gap compared to HSDN, which performs hypergraph-structure level disentanglement, and also outperforms AllSetTransformer by approximately 5 percentage points. These results suggest that naturality-based disentanglement can be effectively applied not only in biomedical domain but also in other domains.
>
>
>
>
> [1] DLGNet: Hyperedge Classification through Directed Line Graphs for Chemical Reactions (OpenReview)

---

### Official Review · Reviewer_8x4v · 2025-07-05

**Clarity:** 3
**Significance:** 3
**Originality:** 3
**Rating:** 4
**Confidence:** 4

**Summary:**

The authors propose a novel method for hyperedge disentanglement through the lens of category theory, where a criterion called factor representation consistency derived from the naturality condition. Hyperedge disentanglement assumes that group interactions have mechanisms and semantics. So this consistency criterion and the leraned representations should remain consistent no matter when the message passing through the direct message passing or disentanglement. To validate the theory and assumption, they create a proof-of-concept model, Natural-HNN, and test on a cancer subtype classification task using genetic pathway data, where genes are nodes and pathways are hyperedges.

**Questions:**

Q1. The current evaluation focuses on hypergraph classification within the bioinformatics domain. Could the authors comment on the model's performance and applicability in different domains? While Appendix D provides results on other datasets, a discussion on why the performance gains are most pronounced in the bioinformatics setting would strengthen the paper's claims about where this criterion is most effective.

Q2. Figure 4 illustrates how the model generates two pathway representations, i.e., $h_{v_i}$ and $y_{v_i}$. Could the difference between these representations like $h_{v_i}-y_{v_i}$ (or difference before the concatenation) be leveraged to directly estimate the disentanglement or consistency of a hyperedge? Such a metric might offer a more straightforward validation of the core mechanism than the more complex analysis presented in the Appendix B.5.

Q3. Can we estiamte the efficiency of learning by providing different numbers of training graphs or training graph sizes? Since graph neural networks can be prone to overfitting, demonstrating strong performance with limited data (e.g., by varying the number or size of training graphs) would further highlight the effectiveness and robustness of the proposed criterion.

**Ethical Concerns:**

["NO or VERY MINOR ethics concerns only"]

**Limitations:**

yes

**Quality:**

3

**Strengths And Weaknesses:**

**Strengths**:

S1. **Well-structured**: The paper is well-written and logically structued. Figure 1 provides an intuitive explanation of the core factor representation consistency criterion, but there is no limitation but ask the neural networks to learn the structural semantics automatically. Many hyper-graph based method is struggle to determine the hyper-parameters to enforce the topology captured, but the proposed method believe that the neural network can learn it automatically.

S2. **Novel Contribution**: The most contribution is to prosose the criterion specifically designed for hyperedge disentanglement. This is a significant step forward for the field of hypergraph representation learning in neural networks. The core idea is that if a hyperedge can be disentangled, then the representations learned through two different pathways (one entangled, one disentangled) should be similar. This concept is simple, straightforward, and allows for learning to be easily integrated from these two channels.

S3. **Strong Empirical Results**: Given the simplicity of the learning process, the detailed analysis provided is essential for supporting the underlying idea. The authors conduct thorough qualitative analyses, using visualizations to verify that their model, Natural-HNN, can indeed capture meaningful biological context and patterns from the data.


**Weaknesses**:

W1. **Missing Analysis on Graph Structures**: The datasets used exhibit considerable variance in node degrees and hyperedge degrees (as shown in Table 3 in the Appendix). However, the paper lacks a discussion on how graph density might correlate with the model's performance improvement across the six different datasets. Such an analysis would provide deeper insight into the model's behavior under varying topological conditions.

W2. **Limited Data Scope**: The evaluation is confined to a single data domain (bioinformatics) and task (hypergraph classification). There is only one domain of data for evaluation. Many general graph representation learning methods can outperform hypergraph-specific ones on standard node classification benchmarks (as data used in Appendix D), which may not be the ideal task for validating the unique advantages of hypergraph learning. The inclusion of synthetic datasets, where the "factor representation consistency" can be explicitly controlled, would also be beneficial for a more rigorous validation of the core criterion.

W3. **Gap Between Theory and Representation Analysis**: The paper provides a theoretical analysis to justify the consistency criterion. However, the evaluation does not fully bridge the gap between this theory and the learned distributed representations. The assessment relies on overall performance and visual patterns, but it is difficult to directly verify the theoretical claims from the representations themselves. An evaluation focused on link-level or hyperedge-level prediction tasks could offer more direct and convincing evidence than the current node-based and graph-based evaluations.

---

> ### Author Rebuttal · Authors · 2025-07-31
>
> Thank you for your constructive review.
> # Q1 & W3 & W2
> While our current evaluation focuses on bioinformatics, our model could be broadly applicable to other domains. We acknowledge the reviewer’s concern that the performance improvements on benchmark datasets in Appendix D are relatively modest compared to those observed in bioinformatics tasks. However, we believe this discrepancy primarily stems from differences in the informativeness of the factors (contexts) encoded in hyperedges across datasets. The more strongly a factor is related to the label, the more critical it becomes to capture that factor, which in turn can have a significant impact on performance. While the degree of informativeness is difficult to quantify, it can be reasonably inferred.
>
> Specifically, in co-citation and co-authorship networks used in Appendix D, hyperedges are formed by simply grouping all documents cited by a single paper or all papers authored by a single researcher. While citations between a pair of papers may carry meaningful context (e.g., a specific reason for citation), it is hard to argue that a group of cited documents shares a well-defined or coherent context. Even if we assume some form of interaction context exists in such hyperedges, the connection between this context and the node labels is often unclear. In such cases, even if the model successfully captures contextual information, it may not significantly contribute to improving label prediction performance.
>
> In contrast, in the cancer subtype classification task, pathways are known to carry functional context [2,3] and are closely associated with the labels (i.e., cancer types) [2,4]. To summarize briefly, pathways are associated with specific biological functions (i.e., factors), and whether these functions operate normally—potentially reflected in the factor representations—can determine the type of disease. This indicates that, in this case, the factors are highly relevant to the labels.
>
> To support our claim that the model is much effective in domains where hyperedges contain rich, label-relevant contextual factors, and to demonstrate its broader applicability, we conducted an additional experiment in the chemical domain—another setting where hyperedges encode meaningful interactions. Specifically, we evaluated our model on the Molecule Reaction Dataset-1 [1], where nodes represent molecules and hyperedges represent chemical reactions annotated with reaction types. The task is to perform hyperedge classification by identifying the reaction type corresponding to each set of participating molecules (i.e., each hyperedge).
>
> |                                        | HGNN        | HNHN        | AllDeepSets   | AllSetTransformer | HSDN        | **Natural-HNN (ours)** |
> |----------------------------------------|-------------|-------------|---------------|-------------------|-------------|--------------------|
> | Molecule Reaction Dataset 1 (Macro-F1) | 0.449±0.005 | 0.257±0.008 |  0.493±0.023 |  0.727±0.026     | 0.491±0.023 | **0.773±0.008**        |
>
>
> While time constraints prevented us from evaluating all baselines during the rebuttal period, we highlight that Natural-HNN substantially outperforms HSDN (a model that also performs structure-level disentanglement) and achieves a 5-point improvement over AllSetTransformer. These results reinforce our claim that our method is particularly effective when hyperedges encode rich, informative contexts, even outside the bioinformatics domain.
>
> We also believe that this response directly addresses the reviewer’s concern regarding the limited data scope (W2) —specifically, the critique that the evaluation was confined to a single domain (bioinformatics) and a single task (hypergraph classification). By presenting results on a new domain (chemistry) and a different task (hyperedge classification), we provide concrete evidence of our model’s broader applicability and generalizability.
>
> # Q2
> Two pathway representations are $h_{e_{i}}^k$ and $\tilde{h}_{e_{i}}^k$. The difference between the two representations can be used to assess the degree of consistency. However, this approach is not applicable to the cancer subtype classification task, as there are no explicit ground-truth labels for the functional roles of individual pathways.
>
>
> # Q3
> Regarding the cancer subtype classification task, we evaluated the performance of our model and the baselines while gradually reducing the number of hypergraphs used during training. The results are presented in Figure 7 of Section 5.4. As shown in the figure, our model maintains strong performance even when the training ratio is reduced.
>
> Additionally, in Figure 8(a), we examined whether our model continues to capture the functional context effectively when the number of training hypergraphs is decreased. The figure shows that our model still captures the functional context well, despite having fewer hypergraphs during training.
>
> For the standard benchmark datasets, please refer to Figure 17 in the Appendix F.3. This figure demonstrates that our model performs well even with a small portion of the training set. Furthermore, in Appendix D.3 (Table 8), we evaluated performance with the training set ratio reduced to as low as 5%, and as shown, our model achieved competitive or superior performance under these conditions.
>
> Therefore, these results demonstrate that the proposed naturality-based criterion enhances the model’s robustness by effectively capturing the functional context or underlying factors, even under limited data conditions.
>
> # W1
> In the cancer subtype dataset, the goal is to predict each patient’s cancer subtype. All patients share the same set of genes and pathways, but their gene features differ. As a result, although this is a hypergraph classification task, it is not affected by topological differences in the hypergraph.
>
>
> [1] DLGNet: Hyperedge Classification through Directed Line Graphs for Chemical Reactions (OpenReview)
>
> [2] Mapping biological process relationships and disease perturbations within a pathway network.
>
> [3] Disentangling the multigenic and pleiotropic nature of molecular function
>
> [4] Identifying cellular cancer mechanisms through pathway-driven data integration

---

> ### Author Response · Authors · 2025-08-07
>
> Dear reviewer 8x4v,
>
> We kindly request you to review our rebuttal, as the author-reviewer discussion period is nearing its end.
> We hope to know whether our rebuttal adequately addressed your concerns. Let us know if you have any remaining questions.

---

### Note · Authors · 2025-08-13

Dear ACs and reviewers,

First, we sincerely appreciate your time spent evaluating our paper and providing such valuable feedback.  We have organized the key points from our discussion with the reviewers for your consideration.

The reviewers have acknowledged several strengths of our work :

* $\textbf{Novel Contribution}$ : Hyperedge disentanglement is an unexplored and novel problem. The naturality concept is simple, straightforward, and allows for learning to be easily integrated. This is a significant step forward for the field of hypergraph representation learning in neural networks.

* $\textbf{Strong Empirical Evidence}$ : The authors conduct thorough qualitative analyses, using visualizations to verify their model. Natural-HNN consistently outperforms other baselines across multiple datasets.

* $\textbf{Presentation}$ : The paper is well-written and logically structued. The paper clearly derives the proposed criterion, which is its key contribution, along with sufficient implementation details.

During the discussion period, we focused on addressing the reviewers’ concerns as follows:

* $\textbf{Additional experiment on chemical reaction dataset}$ : We conducted a hyperedge classification task to demonstrate that our approach is broadly applicable across different domains and task types, and we provided additional explanation on the types of data where it is more effective. (8x4v, gAsC, DgXL)

* $\textbf{Detailed Proof}$ : We provided a more detailed, proof-style explanation of the category theory–based derivation described in the paper to aid understanding. (28h3, DgXL)

* $\textbf{Clarification}$ : We provided explanations to clarify the questions regarding training efficiency and the disentanglement measure. During the discussion with reviewer DgXL, who had given a positive review from the beginning, we included a more formal mathematical description for Q1/W1 in the final comment. (8x4v, gAsC, DgXL)

We deeply appreciate your time and consideration in reviewing these points.

---

### Decision · Program_Chairs · 2025-09-17

**Decision:**

Accept (poster)

**Comment:**

This paper's key strengths include a well-structured derivation of a theoretically grounded loss function, strong empirical performance (4-8% macro-F1 gains over 12 baselines on bioinformatics tasks), and interpretable factor visualizations that align with biological knowledge. However, the work requires further development in several areas: 1) Broaden evaluation beyond biomedical data to include social or synthetic hypergraphs where factor ground truth may be controlled; 2) Strengthen theoretical justification for the naturality criterion and formally define core concepts like "disentanglement" and "factors"; 3) Include structural analysis and task-specific evaluations to better validate the theory; 4) Clarify whether category theory is essential or if simpler methods could achieve similar results. With these improvements, this promising framework could become a cornerstone for hypergraph representation learning.